# Mitochondrial complex I deficiency stratifies idiopathic Parkinson's disease

Irene H. Flønes[1,2,3], Lilah Toker [1,2,3], Dagny Ann Sandnes[1,2], Martina Castelli[1], Sepideh Mostafavi[1,2], Njål Lura[4,5], Omnia Shadad[1,2], Erika Fernandez-Vizarra [6,7], Cèlia Painous[8], Alexandra Pérez-Soriano[8,9], Yaroslau Compta[8], Laura Molina-Porcel [10,11], Guido Alves [12,13], Ole-Bjørn Tysnes [1,2], Christian Dölle [1,2,3], Gonzalo S. Nido [1,2,3] & Charalampos Tzoulis [1,2,3] ✉

Idiopathic Parkinson's disease (iPD) is believed to have a heterogeneous pathophysiology, but molecular disease subtypes have not been identified. Here, we show that iPD can be stratified according to the severity of neuronal respiratory complex I (CI) deficiency, and identify two emerging disease sub-types with distinct molecular and clinical profiles. The CI deficient (CI-PD) subtype accounts for approximately a fourth of all cases, and is characterized by anatomically widespread neuronal CI deficiency, a distinct cell type-specific gene expression profile, increased load of neuronal mtDNA deletions, and a predilection for non-tremor dominant motor phenotypes. In contrast, the non-CI deficient (nCI-PD) subtype exhibits no evidence of mitochondrial impairment outside the dopaminergic substantia nigra and has a predilection for a tremor dominant phenotype. These findings constitute a step towards resolving the biological heterogeneity of iPD with implications for both mechanistic understanding and treatment strategies.

Parkinson's disease (PD) is a heterogeneous, clinicopathologically defined neurodegenerative disorder, with an age-dependent incidence and a rapidly rising global prevalence[1]. More than 95% of affected individuals, in most populations, have idiopathic forms of PD (iPD), i.e., not of monogenic or other known etiology[1,2]. Available treatments for PD provide limited symptomatic relief, while attempts to develop neuroprotective agents have been consistently unsuccessful[3,4].

One of the major challenges in understanding and treating iPD lies in its biological heterogeneity. While all iPD cases exhibit degeneration of the dopaminergic neurons of the substantia nigra pars compacta (SNpc) in the presence of α-synuclein positive Lewy pathology (LP)[5], the disease shows significant clinicopathological variation beyond this common core. Widespread neuronal dysfunction and loss of variable anatomical distribution and severity gives rise to different constellations of motor and non-motor symptoms, with variable rates of

[1]Neuro-SysMed, Department of Neurology, Haukeland University Hospital, 5021 Bergen, Norway. [2]Department of Clinical Medicine, University of Bergen, Pb 7804, 5020 Bergen, Norway. [3]K.G. Jebsen Center for Translational Research in Parkinson's disease, University of Bergen, Pb 7804, 5020 Bergen, Norway. [4]Mohn Medical Imaging and Visualization Centre, Department of Radiology, Haukeland University Hospital, Bergen, Norway. [5]Section for Radiology, Department of Clinical Medicine, University of Bergen, Bergen, Norway. [6]MRC-Mitochondrial Biology Unit, University of Cambridge, Hills Road, Cambridge CB2 0XY, UK. [7]Veneto Institute of Molecular Medicine, 35131 Padova, Italy. [8]Parkinson's disease & Movement Disorders Unit, Neurology Service, Hospital Clínic I Universitari de Barcelona; IDIBAPS, CIBERNED (CB06/05/0018-ISCIII), ERN-RND, Institut Clínic de Neurociències (Maria de Maeztu excellence centre), Universitat de Barcelona, Barcelona, Catalonia, Spain. [9]UParkinson - Sinapsi Neurología, Centre Mèdic Teknon Grup Hospitalari Quirón Salud, Barcelona, Spain. [10]Alzheimer's disease and other cognitive disorders unit. Neurology Service, Hospital Clínic, Institut d'Investigacions Biomediques August Pi i Sunyer (IDIBAPS), Barcelona, Spain. [11]Neurological Tissue Bank, Biobanc-Hospital Clínic-IDIBAPS, Barcelona, Spain. [12]The Norwegian Centre for Movement Disorders and Department of Neurology, Stavanger University Hospital, Pb 8100, 4068 Stavanger, Norway. [13]Department of Mathematics and Natural Sciences, University of Stavanger, 4062 Stavanger, Norway. ✉e-mail: charalampos.tzoulis@uib.no

progression, across affected individuals[1,4,6,7]. This phenotypical diversity has prompted the hypothesis that iPD may encompass multiple distinct disorders, each with its own underlying mechanisms and potential responses to treatments[4,6]. Previous efforts to identify subtypes of iPD have mainly focused on clinical features, with age of onset, and motor or cognitive phenotype being the most common. Several phenotypical subtypes have been proposed based on these classification features, including tremor dominant (TD), akinetic rigid (AR), postural instability and gait disorder (PIGD), and PD dementia (PDD)[8–10]. However, these classifications offer little insight into the pathogenic heterogeneity of iPD. Indeed, top-down approaches attempting to link consensus phenotypical subtypes to specific underlying molecular processes, have thus far been unsuccessful, indicating the need for alternative approaches[4]. It has been proposed that a bottom-up strategy, stratifying iPD according to distinctive molecular features rather than clinical profiles, may prove more fruitful in defining disease subtypes amenable to specific therapies[4].

Major biological processes which have been associated with iPD include mitochondrial dysfunction, impaired proteostasis, and neuroinflammation[1,3,4]. Mitochondrial dysfunction is considered a key feature of iPD, proposed to be integral to its pathogenesis and pathophysiology[11]. Evidence of impaired mitochondrial DNA (mtDNA) maintenance, and respiratory complex I (CI) deficiency has been consistently reported in the dopaminergic SNpc of individuals with iPD[11–14]. While the causes of these mitochondrial defects are unknown, it is hypothesized that they contribute to the neurodegenerative process, based on the facts that chemical CI inhibition[15–17], genetic CI defects[18,19], and mutations in genes regulating mtDNA homeostasis[20,21] can cause degeneration of the dopaminergic SNpc and other neuronal populations in humans. In addition to the SNpc, CI deficiency has been reported in neurons across multiple brain regions, as well as in extraneural tissues, including muscle and blood[12,13]. However, reported findings are variable and contradictory, with nearly half the studies showing no significant difference between individuals with PD and controls at the group level[13]. While this variability may be in part methodological in nature, it also raises a valid question regarding the pervasiveness of CI deficiency in iPD. In line with this notion, studies of CI activity in platelets have shown a large inter-individual spread in PD, with some but not all individuals below the range of controls[22,23] and studies in skin fibroblasts from individuals with iPD have identified evidence of mitochondrial dysfunction in only a minority of the subjects[24,25].

In light of this evidence, we wished to explore whether mitochondrial dysfunction is a universal feature of iPD, or one which occurs in a subpopulation of affected individuals, potentially defining a molecular subtype of the disease. To test this hypothesis, we studied the pervasiveness of neuronal CI deficiency in 92 individuals with genetically validated iPD and 24 neurologically healthy controls from two independent cohorts. We show that iPD can be stratified according to the severity of neuronal CI deficiency, and identify two emerging disease subtypes with distinct molecular and clinical profiles.

## Results

### CI deficiency shows specificity for iPD in the prefrontal cortex
To select a robust marker for neuronal CI immunodetection, we tested antibodies against different CI subunits across the functional domains of the complex[26] in the SNpc of 5 individuals with iPD and 3 controls. All tested subunits exhibited a similar pattern and severity of neuronal CI deficiency, consistent with a quantitative reduction of the entire CI in affected cells (Supplementary Fig. S1). Subunit NDUFS4 was chosen for downstream analyses due to superior staining quality. Neuronal CI status was subsequently assessed, along with the total mitochondrial mass marker VDAC1, in the dopaminergic SNpc and prefrontal cortex (PFC) of two independent cohorts from Norway (NOR, $n = 54$) and Spain (ESP, $n = 62$). The demographic and clinicopathological

information of the cohorts, and experimental allocation of the samples are shown in Table 1 and Supplementary Data 1 and 2.

Neurons were classified as either negative or positive (Fig. 1a, b) by two independent investigators, and the percentage of positive neurons per region and individual was used as a measure of CI status. There was excellent inter-investigator reliability (ICC coefficient = 0.91, 95% C.I (0.87, 0.94), $P = 1.9 \times 10^{-29}$) (Fig. 1c, d, Supplementary Data 3). At the group level, the SNpc exhibited pronounced CI deficiency in both iPD and control subjects, with considerable overlap between the groups. iPD had a significantly lower proportion of CI positive dopaminergic neurons in both cohorts combined ($P = 0.003$) and the NOR cohort ($P = 8.2 \times 10^{-4}$), but not in the ESP cohort ($P = 0.26$) (Fig. 1e, Table 2, Supplementary Data 3). In the PFC, controls showed no or mild CI deficiency, whereas high variation was observed in the iPD group. A significantly lower proportion of CI positive neurons among individuals with iPD was observed in each cohort individually (NOR: $P = 7.0 \times 10^{-4}$, ESP: $P = 0.031$) and in the two cohorts combined ($P = 5.0 \times 10^{-5}$; Fig. 1f, Table 2, Supplementary Data 3). The neuronal CI status in the PFC showed a significant association with disease state (i.e., iPD vs ctr), whereas that of the SNpc did not (linear regression, PFC: $P = 0.006$, SNpc: $P = 0.06$). CI deficiency in the PFC was not associated with post-mortem interval (PMI) ($\rho(109) = -0.08$, $P = 0.42$), age of death ($\rho(110) = 0.03$, $P = 0.73$), or disease duration ($\rho(79) = 0.06$, $P = 0.60$) (Supplementary Data 1). The VDAC1 staining showed no deficient neurons in iPD or controls of either cohort (Fig. 2a, b, Supplementary Data 3).

### Neuronal CI deficiency stratifies iPD
To assess the pervasiveness of CI deficiency in iPD, individuals were stratified based on the proportion of CI positive neurons in the PFC. This region was chosen over the SNpc for two reasons. First, neuronal CI deficiency was found to be more specific to PD in the PFC than in the SNpc (Fig. 1e, f). This is in line with previous findings, showing that dopaminergic SNpc neurons can exhibit severe CI deficiency and other markers of mitochondrial dysfunction with healthy aging, making it challenging to distinguish between disease-specific and age-related changes[27–29]. Second, the severe neuronal loss in the SNpc of iPD[30] complicates the comparison between cases and controls. Specifically, surviving SNpc neurons are likely to be resilient cells, introducing survival bias.

Individuals with iPD were classified by k-means clustering of the NOR and ESP cohorts individually (cases and controls). The number of clusters was set to three based on visual inspection of the data. In each cohort, two subgroups clustered away from the controls and were termed mild and severe CI deficient PD (CI-PD mild; CI-PD severe), whereas one subgroup clustered with the controls and was termed non-CI deficient PD (nCI-PD). When combined, the two CI-PD subgroups accounted for 33% and 22% of the iPD cases, in the NOR and ESP cohorts, respectively. A similar pattern was observed when the combined cohorts were clustered, resulting in three clusters: CI-PD severe ($n = 6/89$, 6.7%), CI-PD mild ($n = 18/89$, 20.2%), and nCI-PD clustering with the controls ($n = 65/89$, 73.0%; Fig. 1g, Supplementary Data 3). Since the severe CI-PD cluster only contained six samples, the mild and severe clusters were combined into a single CI-PD group for downstream statistical analyses.

### CI-PD is characterized by anatomically widespread CI deficiency
To determine the anatomical distribution of neuronal CI deficiency in the iPD subgroups, we studied 15 additional brain regions in representative individuals with CI-PD, nCI-PD and controls from the NOR cohort. All individuals with iPD had Braak scores of 5-6 and the median disease duration was 12 years in both subgroups (Supplementary Data 1 and 2). Subjects in the CI-PD group showed an anatomically widespread neuronal CI-deficiency compared to controls, including the temporal cortex, occipital cortex, CA1 region of the hippocampus,

**Table 1 | Demographic information**

| Variable | Group | n | Median | IQR | U | P-value (Two-sided Mann–Whitney U test) |
|---|---|---|---|---|---|---|
| PMI | NOR: Ctrl | 13 | 43.0 | 31.5–60.0 | 188.0 | 0.16 |
|  | NOR: PD | 39 | 30.0 | 24.0–48.0 |  |  |
|  | ESP: Ctrl | 11 | 7.5 | 6.0–10.3 | 242.0 | 0.48 |
|  | ESP: PD | 51 | 8.0 | 6.0–14.0 |  |  |
|  | Combined: Ctrl | 24 | 27.0 | 7.5–46.8 | 933.5 | 0.31 |
|  | Combined: PD | 90 | 16.0 | 7.0–30.0 |  |  |
| Variable | Group | n | Mean | SD | t | P-value (Two-sided Student's t test) |
| Age of death | NOR: Ctrl | 13 | 82.8 | 10.5 | −1.20 | 0.24 |
|  | NOR: PD | 41 | 79.6 | 7.3 |  |  |
|  | ESP: Ctrl | 11 | 83.5 | 8.9 | −2.10 | 0.038 |
|  | ESP: PD | 51 | 78.0 | 7.4 |  |  |
|  | Combined: Ctrl | 24 | 83.1 | 9.6 | −2.40 | 0.018 |
|  | Combined: PD | 92 | 78.8 | 7.4 |  |  |
| Variable | Group | n |  | OR male (PD/Ctrl) | 95% C.I. | P-value (Two-sided Fisher's exact test) |
| Sex | NOR: Ctrl, male | 4 |  | 3.90 | 1.0–14.9 | 0.056 |
|  | NOR: Ctrl, female | 9 |  |  |  |  |
|  | NOR: PD, male | 26 |  |  |  |  |
|  | NOR: PD, female | 15 |  |  |  |  |
|  | ESP: Ctrl, male | 7 |  | 1.25 | 0.3–4.9 | 0.74 |
|  | ESP: Ctrl, female | 4 |  |  |  |  |
|  | ESP: PD, male | 35 |  |  |  |  |
|  | ESP: PD, female | 16 |  |  |  |  |
|  | Combined: Ctrl, male | 11 |  | 2.33 | 0.9–5.9 | 0.097 |
|  | Combined: Ctrl, female | 13 |  |  |  |  |
|  | Combined: PD, male | 61 |  |  |  |  |
|  | Combined: PD, female | 31 |  |  |  |  |

Demographic information and group comparisons for individuals included in the immunohistochemistry experiments.
*PMI* post-mortem interval, *IQR* interquartile range, *SD* standard deviation, *OR* odds ratio, *C.I.* confidence interval, *NOR* Norwegian cohort, *ESP* Spanish cohort, *PD* Parkinson's disease, *Ctrl* controls.

amygdala, putamen, and the cingulate gyrus. Lower proportions of CI intact neurons were also seen in the internal globus pallidus and the subiculum, but these did not reach statistical significance. No difference from controls was observed in CA4-2 of the hippocampus, the external globus pallidus, inferior olivary nucleus, cerebellar Purkinje cells and the dentate nucleus. The nCI-PD group was similar to controls, with the single exception of the SNpc. The difference in SNpc was, however, less pronounced compared to the CI-PD group. Finally, we noted that, similar to the SNpc, the inferior olivary nucleus, Purkinje neurons and dentate nucleus showed high proportions of CI negative neurons in several of the controls, suggesting that these neuronal populations may be prone to age-dependent CI loss (Figs. 2 and 3, Supplementary Data 3 and 4).

To assess the state of the other mitochondrial respiratory chain (MRC) complexes in the iPD groups, we performed IHC for selected subunits of complexes II-V (CII-V) and VDAC1 in 12 brain regions from two to four subjects from each of the CI-PD and nCI-PD groups (Supplementary Data 2). Irrespective of group, individuals with iPD exhibited no or few neurons staining negative for CIII, CIV and CV (generally ≤5%). No negative neurons were observed in sections stained against VDAC1 or CII (Supplementary Data 3, Supplementary Fig. S2).

### CI-PD is associated with a non-TD phenotype

The CI-PD and nCI-PD groups were similar in terms of age of first reported symptom, age of death, disease duration, and total UPDRS score at the final examination before death. Individuals with severe CI-PD tended to have an earlier age of onset and age of death, but the small size of that group did not allow any conclusions to be drawn (Fig. 4a–c, Supplementary Data 5). The groups differed significantly in the distribution of motor phenotype classified as AR/TD[9] (available

from both cohorts), or PIGD/TD[10] (available from the NOR cohort only), at the time of diagnosis. In the CI-PD group, the AR and PIGD phenotypes accounted for 88% (14/16) and 75% (9/12) of the individuals, respectively. In the nCI-PD group, the AR or PIGD phenotypes were significantly less prevalent, accounting for 60% (25/42) and 29% (6/21) of the individuals, respectively. Notably, individuals with severe CI-PD exhibited exclusively AR or PIGD phenotype (Fig. 4h, i). Furthermore, the CI-PD group exhibited a significantly higher proportion of females (CI-PD: 54.2%, nCI-PD: 26.2%, P = 0.013), compared to the nCI-PD group, and a significantly lower level of ethanol consumption than the nCI-PD group (Fig. 4e, f). There was no significant difference between the groups in the prevalence of dementia at the final examination before death (Fig. 4g). Following multiple testing correction (Benjamini-Hochberg), the difference in the distribution of the PIGD/TD phenotype (P = 0.04), sex (P = 0.04), and number of weekly alcoholic units (P = 0.04) remained significant. However, since the tested variables were highly correlated with each other and the sample size was relatively small, the risk of type 2 error by correcting for multiplicity, is higher than the risk of type 1 error by not correcting[31]. Individual clinical and demographic features are given in Supplementary Data 1 and 5.

In terms of neuropathology, there were no significant between-group differences in either Thal phase staging of beta amyloid, Braak staging of neurofibrillary tangles, or Braak staging of LP. Moreover, there was no difference in the load of LP in the PFC (available from the NOR cohort only) between the groups (Supplementary Fig. S3).

### CI-PD and nCI-PD show different neuronal mtDNA profiles

Neuronal mtDNA copy number and deletion burden were assessed by qPCR in single neurons microdissected from the PFC (Fig. 5a, b) of

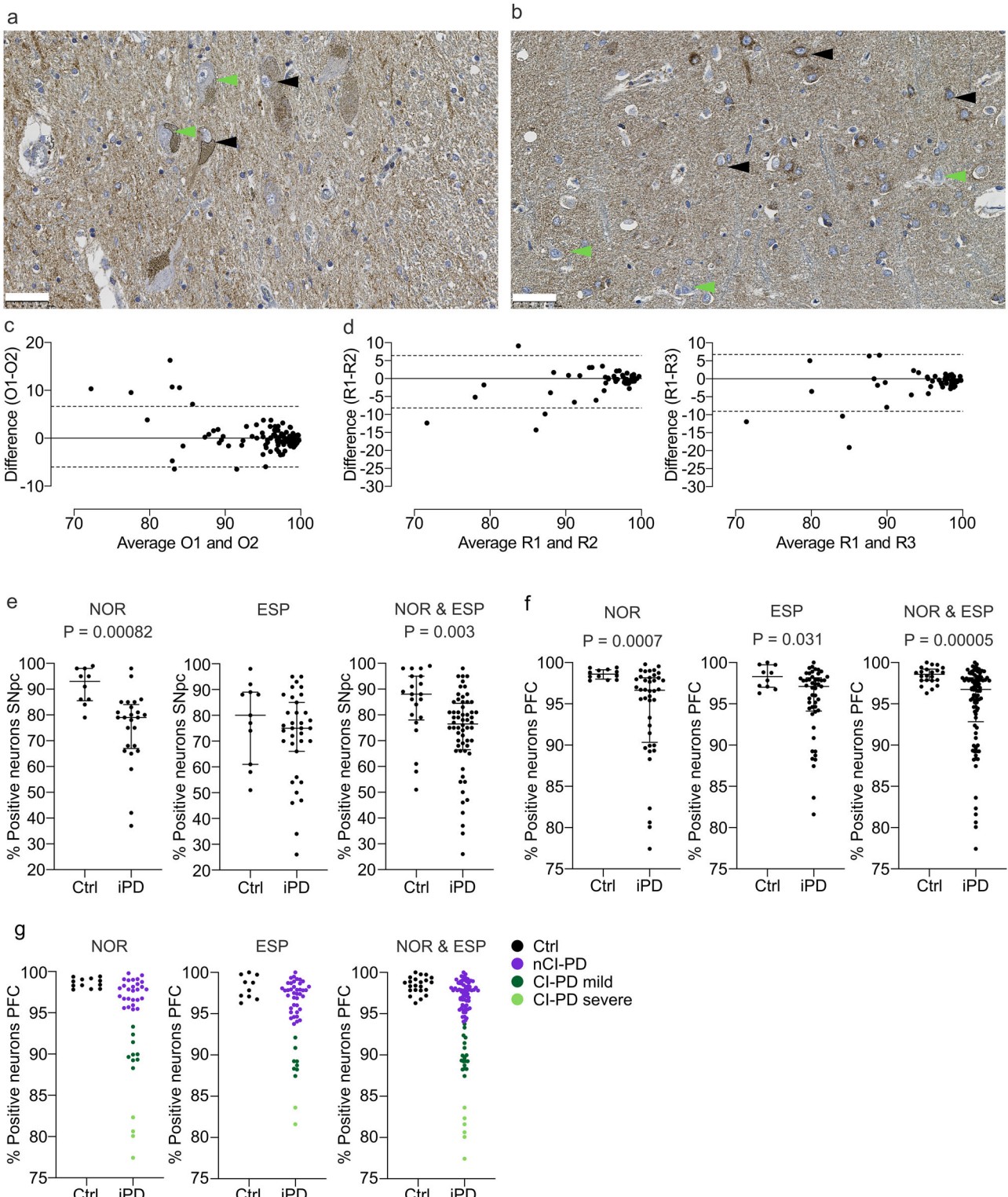

**Fig. 1 | Neuronal CI deficiency stratifies iPD.** SNpc (**a**) and PFC (**b**) sections immunohistochemically stained for CI (NDUFS4). The stippled lines in **a** indicate neuromelanin in dopaminergic neurons of the SNpc. Arrowheads show examples of CI deficient (green) and -intact (black) neurons. Scalebar: 50 μm. **c**, **d** Inter-rater and inter-region replicability of the proportion of CI positive neurons in the PFC. Bland–Altmann plots showing the difference between the measurements of observer 1 (O1) and observer 2 (O2) (**c**), and the difference between measurements by observer 1 in three different regions of the same PFC sections: region 1 (R1) vs region 2 (R2), or region 3 (R3) (**d**). **e**, **f** Scatter plots showing the proportion (in %) of CI positive neurons (y-axes) of controls and individuals with iPD (x-axes) in the

SNpc (controls NOR: $n = 11$, PD NOR: $n = 25$, controls ESP: $n = 11$, PD ESP: $n = 35$; **e**) and PFC (controls NOR: $n = 12$, PD NOR: $n = 40$, controls ESP: $n = 10$, PD ESP: $n = 49$; (**f**). Bars show median and interquartile range. The groups were compared using a two-sided Mann–Whitney U-test. The test statistics are given in Table 2. Multiple comparison adjustments were not made. **g** Scatter plots showing the proportion (in %) of CI positive neurons (y-axes) in the PFC of controls and individuals with iPD (x-axes). K-means clustering classified the iPD subjects into three groups: nCI-PD (purple), CI-PD mild (dark green) and CI-PD severe (light green), in each of the NOR and ESP cohorts and in the combined cohorts.

**Table 2 | Between group comparisons of percentage CI positive neurons**

| Tissue | Cohort | Group | $n$ | Median | IQR | U-value | P-value (Two-sided Mann–Whitney U) |
|---|---|---|---|---|---|---|---|
| SNpc | NOR | iPD | 25 | 78.6 | 67.1–83.9 | 44.5 | $8.2 \times 10^{-4}$ |
| | NOR | Ctrl | 11 | 90.8 | 83.6–97.8 | | |
| | ESP | iPD | 35 | 75.0 | 65.7–87.1 | 148.5 | 0.26 |
| | ESP | Ctrl | 11 | 80.1 | 60.7–89.4 | | |
| | Combined | iPD | 60 | 76.3 | 66.7–84.2 | 372.5 | 0.003 |
| | Combined | Ctrl | 22 | 87.1 | 77.1–94.7 | | |
| PFC | NOR | iPD | 40 | 96.6 | 90.3–98.1 | 84.0 | $7.0 \times 10^{-4}$ |
| | NOR | Ctrl | 12 | 98.6 | 98.0–99.1 | | |
| | ESP | iPD | 49 | 97.1 | 94.1–98.1 | 138.0 | 0.031 |
| | ESP | Ctrl | 10 | 98.3 | 97.0–99.7 | | |
| | Combined | iPD | 89 | 96.8 | 92.8–98.1 | 431.0 | $5.0 \times 10^{-5}$ |
| | Combined | Ctrl | 22 | 98.6 | 97.8–99.2 | | |

*IQR* interquartile range, *SNpc* substantia nigra pars compacta, *PFC* prefrontal cortex, *NOR* Norwegian cohort, *ESP* Spanish cohort, *iPD* idiopathic Parkinson's disease, *Ctrl* controls.

individuals with CI-PD ($n = 6$), nCI-PD ($n = 9$) and controls ($n = 7$; Supplementary Data 2). Eight to sixteen single neurons were assessed per individual. There was a significant difference in neuronal mtDNA deletions between the groups (ANOVA, $P = 0.037$, $\eta^2 = 0.30$). Post-hoc analyses showed that the CI-PD group had significantly higher levels of mtDNA deletions compared to the nCI-PD group ($P = 0.037$; Fig. 5b). No differences were observed in neuronal mtDNA copy number (total or non-deleted) between the groups (Fig. 5c, d). The mtDNA data is shown in Supplementary Data 6.

### CI-PD and nCI-PD exhibit distinct transcriptomic profiles

Hierarchical clustering of RNA-seq data from the PFC of 98 of the individuals previously assessed for CI deficiency (Supplementary Data 2) indicated that individuals with CI-PD tended to cluster together (Fig. 6a), with a similar behavior observed in each of the cohorts (Supplementary Fig. S4). The CI-PD group was associated with lower DV200 values, a measure of RNA integrity indicating the percentage of RNA fragments with 200 nucleotides or more, compared to both nCI-PD and controls (Fig. 6a). However, this association could not be explained by PMI, since the PMI range of the CI-PD group was similar to that of the controls within each cohort (Supplementary Fig. S5).

To control for between-group differences in the cell composition of the samples, we assessed Marker Gene Profiles (MGPs)[32] of different cell types in each iPD group and the controls. Linear regression adjusting for age of death, sex, PMI and cohort, indicated a significant change in the MGP of multiple cell types in the CI-PD group compared to the controls. However, step-wise regression analysis suggested that the observed changes were not independent, but rather driven by a decrease in the MGP of GABAergic interneurons expressing both vasoactive intestinal peptide-expressing (VIP) and reelin (GabaVIPReln) (Fig. 6b, Supplementary Fig. S6). No difference was observed in the MGPs of the nCI-PD group compared to the controls (Fig. 6b, Supplementary Fig. S6).

To account for the effects of different cell composition on the differential gene expression (DE) profile of CI-PD, we performed DE analysis using two alternative models: 1) Correcting for the confounding MGP of GabaVIPReln (Model_1) and 2) not correcting for MGPs (Model_2). Both models included the demographic variables and oligodendrocyte MGPs to adjust for any variation in the gray/white matter content between the samples[33,34]. Adjusting for GabaVIPReln MGPs resulted in a striking reduction of the number of differentially expressed genes in the CI-PD group, from 11,138 to 10 (Supplementary Data 7). Notably, oxidative phosphorylation (OXPHOS)-related genes were among the top differentially downregulated genes in CI-PD, in the non-adjusted model. However, none of these genes remained

significant after adjusting for cell composition (Fig. 6c). In contrast, MGP correction had no major impact on the DE analysis of the nCI-PD group (Fig. 6c).

### nCI-PD exhibits altered expression in multiple PD-linked pathways but no decrease in OXPHOS genes

We next performed functional enrichment analyses of the DE results. Since the DE analysis of the CI-PD group was highly confounded by the underlying tissue composition, we were unable to determine whether the results represented true regulatory changes or simply altered cell composition. We therefore chose not to further focus on these results, which are given in Supplementary Data 8. The transcriptomic signature of nCI-PD, which was not confounded by estimated cell composition, was characterized by changes in numerous pathways, including the downregulation of processes related to proteostasis (lysosome and proteasome), GABAergic signaling, synaptic function and lipid metabolism. Upregulated processes included pathways related to epigenomic regulation, steroid synthesis, and potassium channels. Notably, there were no changes in genes or pathways related to mitochondrial respiration (Fig. 6d, Supplementary Figs. S7 and S8, Supplementary Data 8).

### Single-nuclei RNA-seq shows distinct cell type-specific profiles in the iPD subtypes

To mitigate the bias of cell composition indicated by the bulk RNA-seq data and assess cell type-specific transcriptomic similarities between CI-PD and nCI-PD, we performed massively parallel single-nuclei RNA-seq (snRNA-seq) in fresh-frozen PFC from $n = 7$ individuals with CI-PD, $n = 5$ individuals with nCI-PD, and $n = 6$ controls (Supplementary Data 2). After filtering and quality control, a total of 117,105 nuclei (mean: $6506 \pm 2952$ nuclei per individual) were included in the analyses. There was no significant difference in the number of captured nuclei between groups (ANOVA: F = 3.65, degrees of freedom = 2, $P = 0.051$).

Default clustering using Seurat identified a total of 28 clusters that could be unequivocally assigned to a major cortical cell type, based on established cell type markers (12 excitatory neuron clusters, 8 inhibitory neuron clusters, 3 oligodendrocyte clusters, 2 astrocyte clusters, 1 microglia cluster, 1 oligodendrocyte precursor cells (OPC) cluster, and 1 endothelial cluster, Fig. 7a, b). Unique molecular identifier (UMI) counts and number of identified genes per nucleus were not significantly different between groups in any of the identified cell type clusters.

Differential gene expression analysis comparing each of the iPD groups to the controls revealed distinct cell type-specific transcriptomic profiles between CI-PD and nCI-PD. After FDR correction,

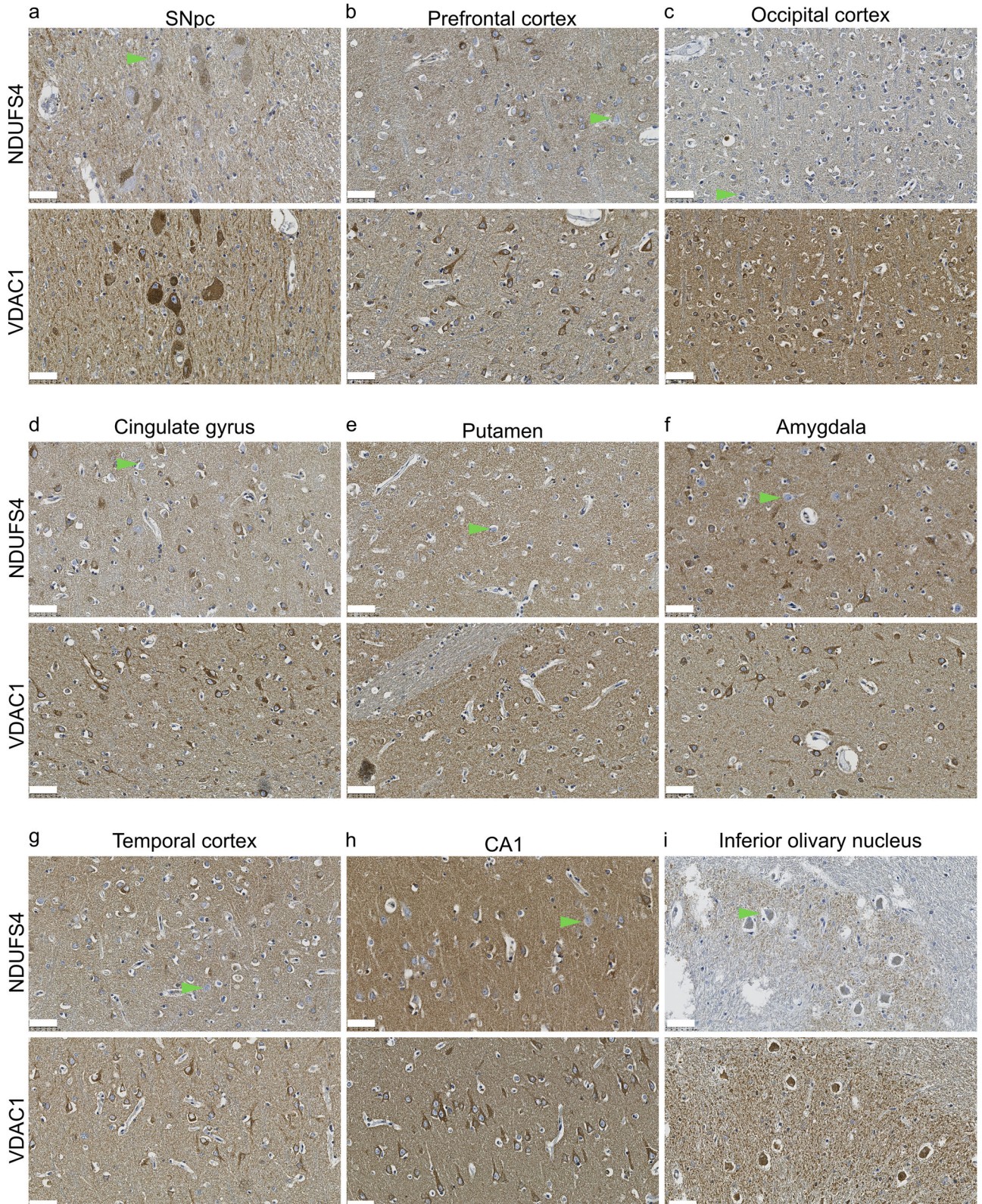

**Fig. 2 | CI and VDAC1 immunostaining in selected brain regions.** Representative images of immunostaining against CI (NDUFS4) and VDAC1, in areas showing CI deficiency in iPD; substantia nigra pars compacta (SNpc; **a**), prefrontal cortex (**b**), occipital cortex (**c**), cingulate gyrus (**d**), putamen (**e**), amygdala (**f**), temporal cortex (**g**), CA1 of the hippocampus (**h**) and inferior olivary nucleus (**i**). Green arrowheads show examples of CI negative neurons. No negative neurons were observed in the VDAC1 staining. Scalebar: 50 μm.

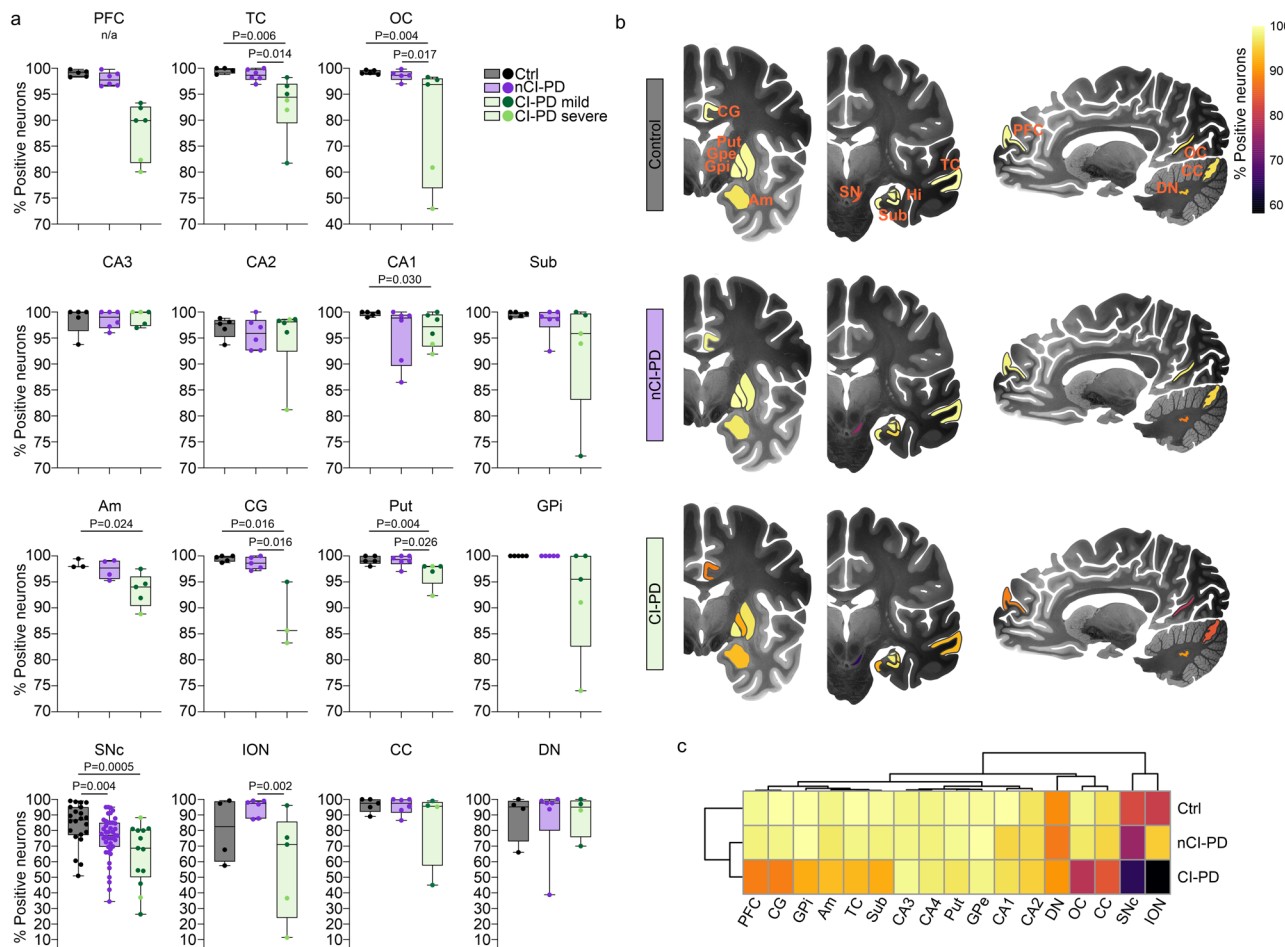

**Fig. 3 | Anatomical distribution of neuronal CI deficiency in the nCI-PD and CI-PD groups. a** Box plots showing the proportion (in %) of CI positive neurons in 16 brain regions from controls (gray), and individuals with nCI-PD (purple) and CI-PD (green). The boxplot from PFC represents the spread in all individuals, dots show individuals included in downstream analyses for regional CI staining. Box plots show individual values (dots), median and interquartile range (box) and minimum-maximum range (whiskers). Statistical analysis was not performed in the PFC, since this region was used to classify the samples, and the samples were selected to represent the full spectrum of CI status for each group. Nominal p-values of two-sided Mann-Whitney U-tests are shown. Multiple testing correction was not performed for these analysis, due to the relatively small sample size and high correlation between the CI state of different anatomical regions per individual, as

illustrated in the heatmap (**c**). All individuals within the CI-PD group are shown in the same green boxplot, but differentiated by the color of the points into mild (dark green) and severe (light green) CI-PD. **b** Structural brain maps illustrating the anatomical distribution of neuronal CI deficiency in the control, nCI-PD and CI-PD groups. The median proportion of CI positive neurons is shown as a heatmap. Brain illustrations are modified images from the 7MRI atlas published by Edlow et al.[91]. **c** Hierarchical clustering of the median proportion of CI positive neurons in each group. SNpc substantia nigra pars compacta, PFC prefrontal cortex, OC occipital cortex, TC temporal cortex, Sub subiculum, Am amygdala, CG cingulate gyrus, Put putamen, GPe external globus pallidus, GPi internal globus pallidus, ION inferior olivary nucleus, CC cerebellar cortex, DN dentate nucleus, Hi hippocampal regions CA4-CA1, Ctrl controls. n/a: statistical comparison not performed.

we found a total of 148 significantly differentially expressed genes in CI-PD and 466 significantly differentially expressed genes in nCI-PD, distributed among several cell type clusters, with 47 genes overlapping between the groups (Fig. 7c). CI-PD showed differential expression in most excitatory and inhibitory neuronal subtypes, and a notable upregulation in microglia, including *PRKN* – encoding parkin, a key component of mitophagy – in which mutations cause autosomal recessive PD[35]. nCI-PD exhibited multiple differentially expressed genes, with most of the signal observed in neurons and oligodendrocytes, while no differential expression was seen in microglia. The differentially expressed genes per cell type and group are shown in Supplementary Data 9 and 10. A visual map of significantly differentially expressed genes (FDR < 5%) with fold change ≥25%, in at least one cell type, is shown in Supplementary Fig. S10.

To directly identify features differentiating CI-PD and nCI-PD, we performed DE between the two groups. In this contrast, we identified a total of 909 genes differentially expressed in at least one cell type, suggesting that differences between the iPD subtypes were larger than

between each subtype and the controls. We, therefore, focused further analyses on this contrast. Significant differential expression was found across all cell types, neuronal and glia, with the highest number of differentially expressed genes seen in excitatory neuron clusters ex_09 and ex_19. According to the expression of known neuronal markers[36], ex_09 are located in cortical layers L4c/L6, and express relatively high levels of *OPRK*, encoding kappa opioid receptors. These neurons are, therefore, potentially involved in nociception. Neurons classified as ex_19 are located in layers (L6/L6b), and have a high relative expression of *ADRA2A*, encoding the alpha 2A adrenoreceptor, consistent with adrenergic afferentation.

Most differentially expressed genes were specific to 1–2 cell types, with few notable exceptions, including *SRSF5*, *MAK*, and *RAB11A* which were downregulated in CI-PD across several excitatory and inhibitory neurons, and/or glia. A focused analysis of the nuclear genes encoding MRC proteins revealed a general trend for downregulation in CI-PD, compared to nCI-PD, including significantly lower expression of multiple subunits of complexes I, III, IV, and V. Similar to the overall

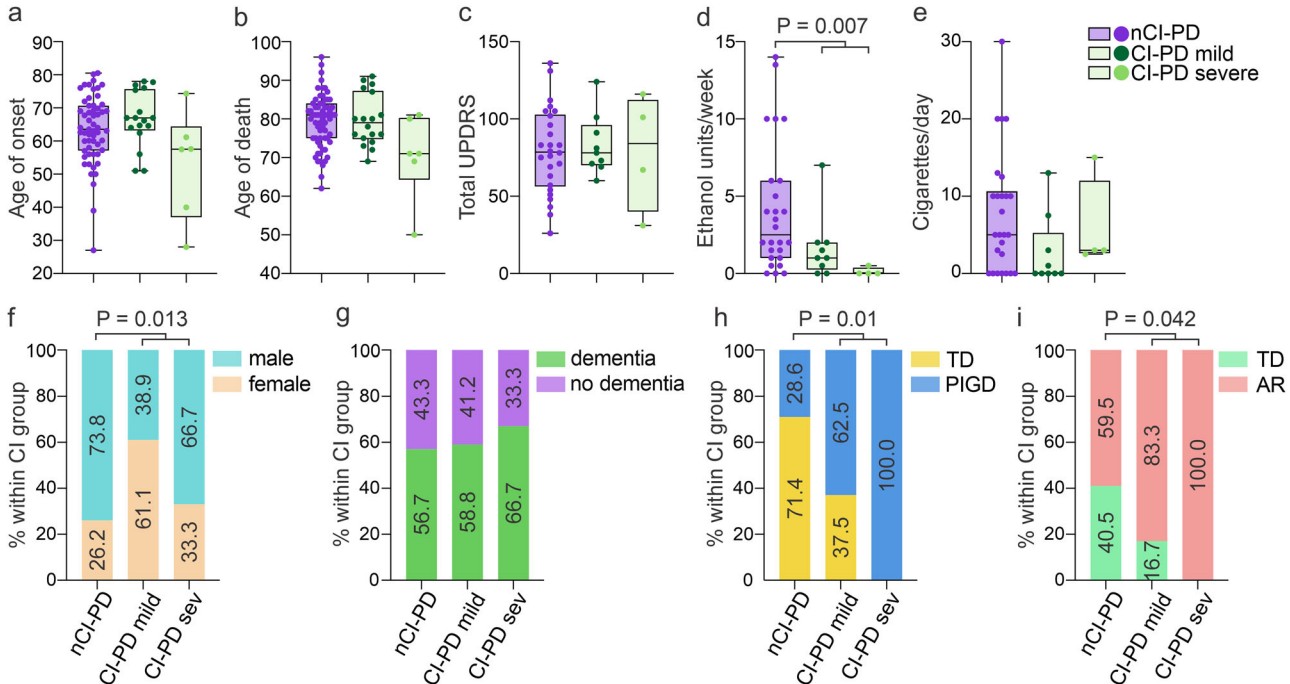

**Fig. 4 | Clinical differences between CI-PD and nCI-PD.** Box plots (**a**–**e**) show individual values (dots), median and interquartile range (box) and minimum-maximum range (whiskers) of age of onset (**a**), age of death (**b**), total UPDRS at final clinical examination (**c**), total number of weekly ethanol units at the time of diagnosis (**d**), and total number of daily cigarettes at the time of diagnosis (**e**) for the three iPD groups (nCI-PD: purple, CI-PD mild: dark green, CI-PD severe: light green). Statistical significance was tested between nCI-PD and the entire CI-PD group (i.e., not differentiating between mild and severe), due to the low number of individuals in the severe group, using the two-sided Mann–Whitney U-test. Bar plots (**f**–**i**) show for each iPD subgroup the distribution (in percentage) of males/

females (**f**), presence of dementia at final clinical examination (**g**), PIGD/TD phenotype (**h**) and AR/TD phenotype (**i**), at the time of diagnosis within each iPD group. Statistical significance was tested using the $X^2$ test. Demographic data, data on dementia at time of death, and the AR/TD-clinical phenotypes were available from both cohorts. All other clinical variables were available only from the NOR cohort. The number of included individuals is not equal in the different comparisons and is shown in Supplementary Data 5. AR akinetic rigid, TD tremor dominant, PIGD postural instability and gait disorder, UPDRS Unified Parkinson's Disease Rating Scale.

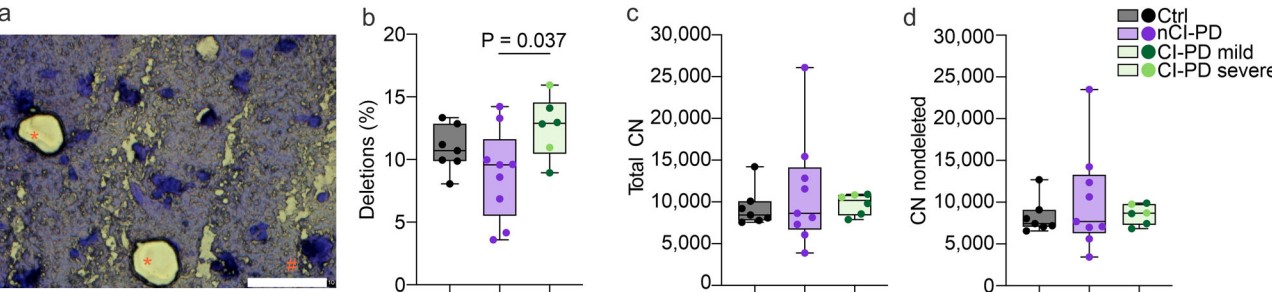

**Fig. 5 | The nCI-PD and CI-PD groups show different neuronal mtDNA profiles.** Single neurons were collected from frozen PFC sections using laser-microdissection and mtDNA was analyzed for copy number (CN) and deletion proportion using qPCR. Eight to sixteen single neurons were assessed per individual from individuals with CI-PD (n = 6), nCI-PD (n = 9) and controls (n = 7). **a** Microphotograph of a representative PFC section, stained with cresyl violet and

showing neurons before (#) and after (*) microdissection. Scale bar 10 μm. **b** Neuronal mtDNA deletion levels. **c** Total neuronal mtDNA copy number. **d** Non-deleted neuronal mtDNA copy number. Box plots (**b**–**c**) show median and interquartile range (box) and minimum-maximum range (whiskers) of the mean values for each individual (points) per group. Statistical significance of group difference was tested using two-sided ANOVA. CN copy number, Ctrl control.

differential expression signature, these changes were predominantly seen in neuronal clusters ex_09 and ex_19 (Supplementary Fig. S11). Transcriptional differences in MRC genes between either iPD subtype and controls did not reach statistical significance. Nevertheless, there was a notable trend for upregulation in nCI-PD (Supplementary Figs. S12 and S13).

Functional enrichment of the differentially expressed genes was carried out independently for each cell type cluster by overrepresentation analysis of the Kyoto Encyclopedia of Genes and Genomes (KEGG) pathways. Most cell types showed no significant

enrichment, with the exception of ex_19 which showed significant enrichment for pathways relating to OXPHOS, the ribosome, and Parkinson's and Huntington's disease (the latter two also driven by OXPHOS genes). Additionally, the ribosome pathway was enriched in clusters ex_09 and ex_24 (Supplementary Data 12). Finally, we performed overrepresentation analysis on the union of significantly differentially expressed genes (FDR < 5%) in excitatory neurons, inhibitory neurons, and glia. This analysis showed a highly significant enrichment in OXPHOS- and ribosome related pathways in excitatory neurons (Supplementary Data 13).

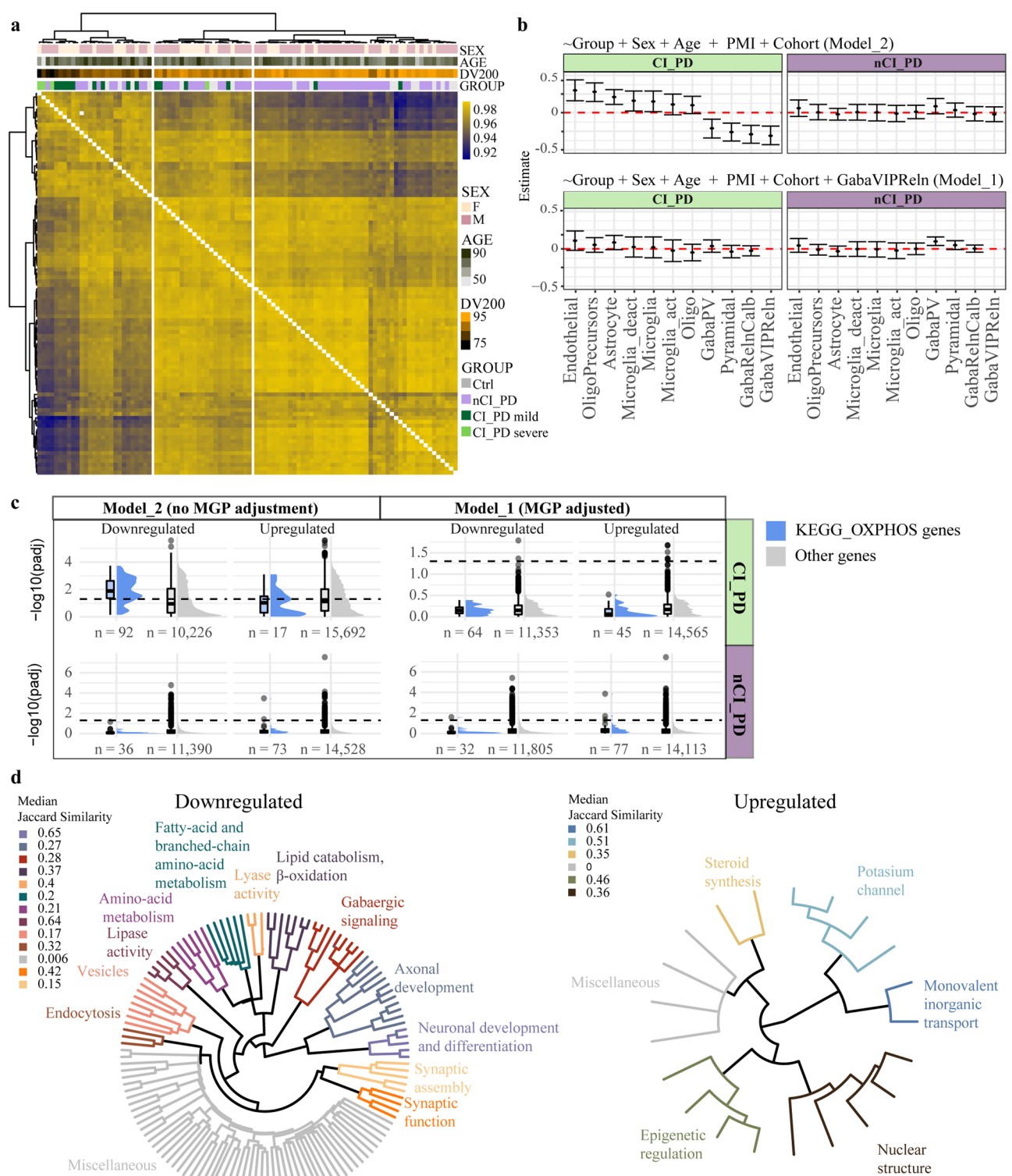

**Fig. 6 | nCI-PD and CI-PD groups exhibit distinct transcriptomic profiles in the PFC. a** Hierarchical clustering of the samples based on sample-sample correlation. The heatmap colors indicate Pearson's correlation values between each pair of samples. AGE: age of death. **b** The estimated group difference (y-axis) in MGPs of the cell types (x-axis) based on linear regression using the two different models are shown. The cell types are described in Mancarci et al.[66]. The estimate and the 95% confidence intervals are indicated by the point and the whiskers, respectively (**c**) Significance (shown as -log10 of the adjusted *p*-value) of genes included in the KEGG OXIDATIVE_PHOSPHORYLATION pathways (blue) or all other genes (gray) among the downregulated/upregulated genes based on a model not adjusting (Model_2) or adjusting (Model_1) for the confounded MGPs, in each of the iPD subtypes. Dashed line indicates adjusted pvalue of 0.05. Boxplots represent the interquartile range (IQR) with the median indicated by the bold line. Whiskers are extensions from the top and the bottom of the boxplot by a value of 1.5*IQR. Values outside of the whisker range are indicated by circles. **d** Visualization of the enrichment analysis outcome of the nCI-PD group, based on DE analysis using Model_2. Each leaf of the tree represents one significantly enriched gene-set. Jaccard similarity was used to cluster the gene-sets based on semantic similarity. Each cluster was annotated to represent the main function represented by its members. The names of the specific gene-sets are shown in Supplementary Fig. S7 (downregulated) and Supplementary Fig. S8 (upregulated).

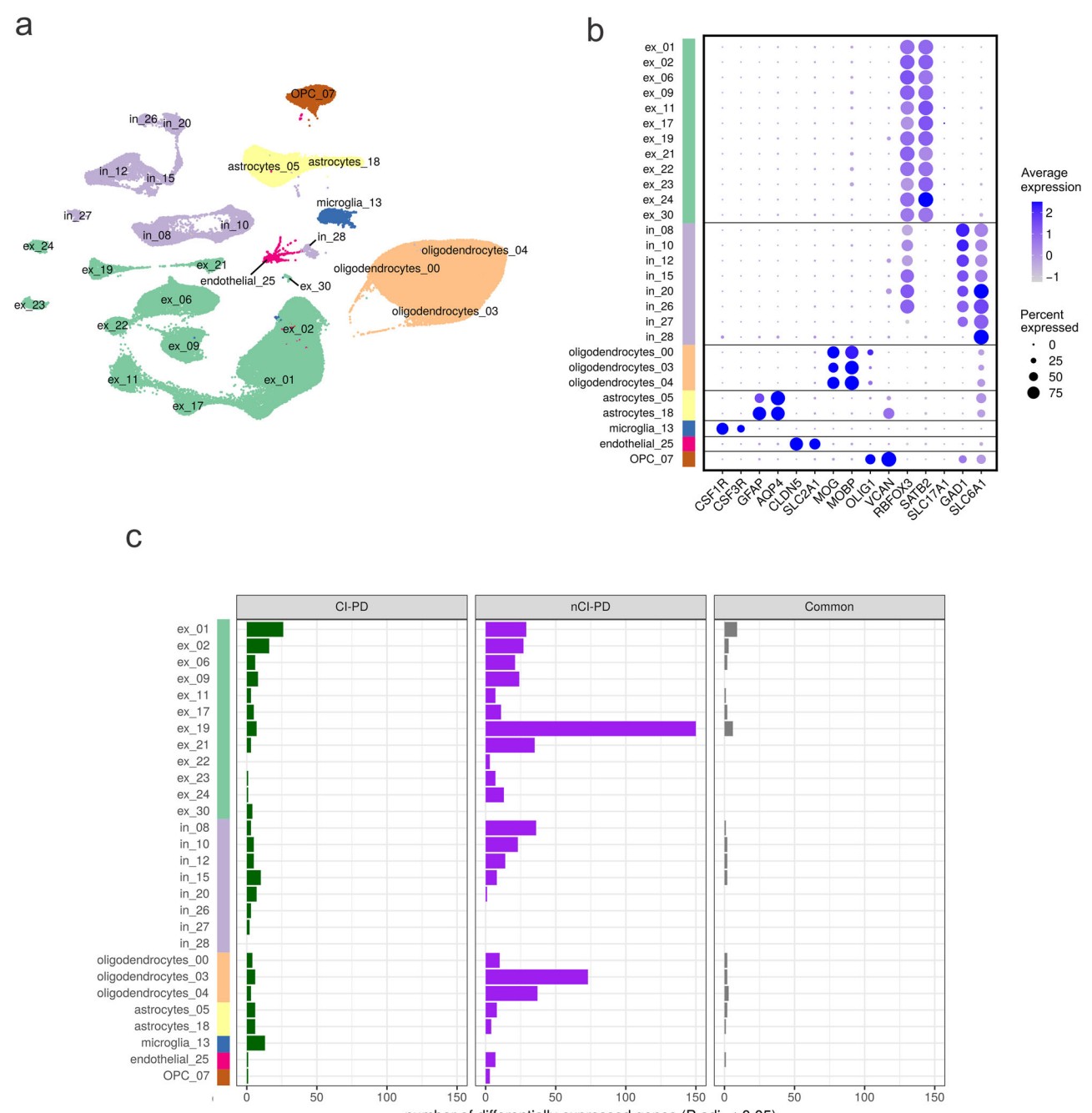

**Fig. 7 | Single-nuclei RNA-seq shows distinct cell type-specific transcriptional profiles for CI-PD and nCI-PD. a** UMAP plot of the filtered snRNA-seq dataset (117,105 nuclei) showing the clustering in the main cortical cell types (ex: excitatory neurons, in: inhibitory neurons, oligodendrocytes, astrocytes, microglia, OPCs, and endothelial cells). **b** Dotplot showing the average cell type markers' expression in each of the 28 subclusters identified by Seurat, unequivocally mapping the nuclei to the main cortical known cell types. The color of the dots indicates average scaled expression, the size representing the proportion of nuclei in the cluster that express the marker. **c** Summary of the number of differentially expressed genes in autosomes (at FDR < 5%) within each cluster for the CI-PD *vs* controls (left hand side panel, green bars), for the nCI-PD *vs* controls (center panel, purple bars), and common between the two contrasts (right hand side panel, gray bars).

## Discussion

Identifying and exploring molecular subtypes of iPD is a critical step towards mechanistic and therapeutic advancements. We show that iPD can be stratified according to the severity of neuronal CI deficiency and identify two emerging disease subtypes with distinct molecular and clinical features, indicating a divergent pathophysiology. CI-PD is characterized by widespread neuronal CI deficiency across multiple brain regions. In contrast, nCI-PD, exhibits a neuronal CI profile similar to that of demographically matched controls, with the single exception of the dopaminergic SNpc, where higher levels of CI deficiency are

observed, albeit to a lesser extent than in CI-PD. Importantly, this difference is not driven by disease duration or age, indicating that it does not reflect two different stages of the same process. The existence of the CI-PD and nCI-PD subtypes is shown independently in two different populations, suggesting that this is generalizable to- and representative of iPD.

It is likely that the observed CI deficiency contributes to neuronal dysfunction and death in CI-PD. Neurons depend on CI function to maintain a high ATP turnover[37]. Brain mitochondria exhibit impaired energy metabolism with major decreases in ATP production upon 60%

reduction of CI activity, and this threshold is much lower (~25%) for synaptic mitochondria[38]. While functional assessment of CI activity is currently not feasible in individual neurons from postmortem brain, it is reasonable to assume that neurons with no detectable CI staining (i.e., CI negative neurons) in our samples must have substantially decreased CI levels and, therefore, a compromised bioenergetic status. This is further corroborated by the fact that isolated CI deficiency of either chemical[15,17] or genetic[18,39] etiology is an established cause of neuronal dysfunction and death in humans and other animals.

With the single exception of mild deficiency in the dopaminergic SNpc, the nCI-PD group exhibited a neuronal CI profile similar to that of age-matched controls. Thus, we postulate that CI deficiency does not make an important contribution to the pathogenesis of PD-specific neurodegeneration outside the SNpc in these individuals. CI deficiency may still contribute to the degeneration of the dopaminergic SNpc neurons in nCI-PD. However, given that this deficiency was significantly milder compared to CI-PD, and showed substantial overlap with the aged-matched controls, its disease-specific pathophysiological significance remains uncertain.

The CI-PD and nCI-PD groups differed in terms of motor disease subtype at the time of diagnosis. Our data indicate that CI dysfunction is significantly more common in the non-TD form of iPD, which has also been shown to harbor more severe neurodegeneration[40]. The mechanism underlying this association is currently unknown but may be related to more widespread and severe neuronal dysfunction and loss in CI-PD, induced by the CI deficiency. This is corroborated by the bulk RNA-seq data indicating more pronounced neuronal loss in CI-PD. Until clinically applicable biomarkers of CI-PD and nCI-PD can be developed, enabling us to classify individuals in vivo, perhaps clinical studies could benefit by distinguishing between TD and non-TD phenotypes. Since individuals with a non-TD phenotype at the time of diagnosis are substantially more likely to belong to the CI-PD subtype, they may be more likely to benefit from experimental treatments targeting mitochondria. While there are currently no drugs specifically targeting CI, several agents targeting mitochondrial biogenesis and/or function have been tested in clinical trials with negative results[3], while others are still in the pipeline[41]. In light of our findings, it would be of interest to revisit the data of previously conducted trials and assess how the treatment impacted individuals with a non-TD or TD phenotype. In addition to the motor phenotype, we observed an association of CI-PD with the female sex and a lower consumption of ethanol. The latter may be driven by the female preponderance in this subtype.

Neuronal mtDNA profiles in the PFC corroborated the stratification of iPD into CI-PD and nCI-PD groups. mtDNA deletions are an established cause of MRC deficiencies in post-mitotic cells[42–44]. However, given the small effect size of the mtDNA deletion increase in the CI-PD group, and the fact that neuronal deletion levels were generally below 60%, i.e., the threshold at which MRC dysfunction has been proposed to occur[42], the biological significance of the detected difference in mtDNA deletion levels remains uncertain. It should be noted that, since most PFC neurons of individuals with CI-PD remain CI positive, it is possible that few or none of the captured neurons in our experiments were CI negative. Thus, the possibility that CI negative neurons harbor higher deletion levels, driving the CI deficiency, cannot be ruled out by this work. An alternative explanation is that mtDNA deletions in CI-PD individuals may be the result, rather than the cause, of CI deficiency. Under this hypothesis, mtDNA deletions may arise during the repair of double-strand breaks, caused by increased reactive oxygen species (ROS) generation in CI deficient neurons[45,46].

RNA-seq revealed distinct transcriptomic profiles between the CI-PD and nCI-PD groups. In bulk tissue data, the transcriptomic signature of CI-PD exhibited downregulation of OXPHOS-related genes but was also severely confounded with cell-type-specific expression profiles (i.e., MGPs), indicating differences in the cell type composition of the samples. The observation that CI-PD exhibited decreased estimates for GABAergic interneurons is consistent with a mitochondrial defect. GABAergic interneurons have high bioenergetic demands, due to their fast-spiking firing pattern, and have been shown to be selectively lost in mitochondrial disease characterized by neuronal CI deficiency disease[47]. While MGPs are not direct cell counts, they are indicative of a simultaneous shift in multiple cell-type specific genes. Regardless of whether this reflects a difference in the underlying cellular composition of the tissue, synaptic loss, decreased transcriptional output, or phenotypic change[39], this association indicates that any regulatory changes cannot be disentangled from differences due to altered tissue composition[33,48]. Thus, no further conclusions should be drawn from transcriptomic analyses of bulk brain tissue data from CI-PD individuals. In contrast, the nCI-PD group exhibited no difference in MGPs compared to controls. Thus, we believe that the DE signature of this group reflects regulatory changes associated with the disease. The transcriptional profile of nCI-PD showed altered expression in multiple pathways, many of which have been previously linked to PD[49–51].

snRNA-seq allowed us to compare the iPD subgroups within each cell-type, thus circumventing the confounder of different cell composition. These analyses revealed a highly distinct cell type-specific transcriptomic profile for each of CI-PD and nCI-PD, supporting that the two subtypes harbor distinct molecular processes. While single-nucleus transcriptomics are not sufficient to adequately characterize these processes, since they only capture a small part of the cell's transcriptional state, the data provides us with some hints. Neurons and astrocytes were affected in both groups, albeit by largely different genes. However, microglia and oligodendrocytes showed subgroup-specific involvement. The CI-PD-specific changes in microglia are suggestive of mitophagy activation, which indicates the presence of mitochondrial dysfunction and stress in these cells[52]. In contrast, nCI-PD showed no changes in microglia, but harbored substantial transcriptional alterations in oligodendrocytes. Oligodendrocyte involvement has been previously reported in PD[53]. Our findings suggest that this may be a feature of nCI-PD.

Direct comparison of the CI-PD and nCI-PD groups further highlighted their distinct molecular profiles, as it revealed substantially more differentially expressed genes than in each of the subtypes compared to controls. At the single-nucleus level, CI-PD was characterized by a significantly lower expression of MRC-encoding genes compared to nCI-PD. This difference was largely driven by an upregulation trend in nCI-PD, compared to controls. One possible interpretation of this observation is that, in the majority of individuals with PD (i.e., nCI-PD), neurons exhibit an upregulation of mitochondrial respiration and/or biogenesis, potentially indicative of a compensatory response to disease-related stress. Conversely, individuals with CI-PD may represent cases where this compensatory response fails, possibly due to underlying mitochondrial pathology.

Our findings must be interpreted considering certain limitations. Although we defined subgroups of iPD, the exact threshold between them remains arbitrary, and the severity of neuronal CI deficiency across individuals can also be seen as a continuum. Nevertheless, this does not alter the fact that only a subset of individuals with iPD show neuronal CI deficiency outside the SNpc, and this subpopulation exhibits distinct clinical and molecular features.

Functional assessment of specific CI activity was not undertaken in this work, because it is currently not feasible to assess in individual cells. Measurement of CI activity in bulk tissue would be unreliable because of both low sensitivity due to the mosaic distribution of the CI defect, and low specificity in the sense that it cannot adequately distinguish between cell-specific reduction in CI activity or a decrease in neuronal and synaptic content in the sample. We have previously shown that the PFC of iPD exhibits markers of decreased neuronal content[33], and our current data suggests that these differences may be more pronounced in CI-PD, thereby biasing any comparison in bulk tissue. In terms of the gene expression data, the cell-type composition

bias is mitigated by the snRNA-seq. However, when interpreting this data, it is important to consider that nuclear transcriptomes do not reflect post-transcriptional RNA turnover and subcellular segregation and do not, therefore, necessarily capture the full transcriptomic state of the cell. This is particularly true in neurons, where RNA composition varies considerably between the soma, processes, and synapses. Furthermore, snRNA-seq cannot capture the mitochondrial transcriptome.

In summary, this work establishes that anatomically widespread neuronal CI deficiency is not a pervasive feature of iPD, but one that occurs in a subset of individuals, accounting for approximately a fourth of all cases. These results have important scientific and clinical implications for the field. First, the existence of the CI-PD and nCI-PD subtypes provides an explanation for the variable and conflicting findings in previous studies of mitochondrial pathology in iPD, which were performed on unstratified samples. Second, given the major molecular differences between the two iPD subtypes, we recommend that future studies employing iPD brain tissue should stratify their material according to CI subtype. This would decrease the biological heterogeneity of the sample and allow for more consistent and biologically meaningful observations to be made. Finally, it is plausible that the CI-PD and nCI-PD subtypes exhibit different susceptibility to therapeutic interventions targeting mitochondria or other subtype-associated pathways. Thus, we believe that harnessing this classification in clinical practice may be an essential step towards implementing precision medicine in iPD. While our findings suggest that the non-TD motor phenotype may be used as a proxy for CI-PD to some extent, the accuracy of this approach is limited. To efficiently translate these findings into clinical practice requires the development of clinically applicable biomarkers, allowing us to classify individuals in vivo.

## Methods
### Ethics approval
Ethics approval for this work was obtained from The Regional Committee for Medical and Health Research Ethics, Western Norway (REK Vest; REK 2017/2082, 2010/1700, 2016/1592).

### Subject cohorts and tissue samples
Brain tissue was obtained from four independent cohorts of European origin. The Norwegian cohort (NOR, n = 55) comprised 41 individuals with iPD from the ParkWest study, a prospective population-based cohort followed since 2004[54], and 13 neurologically healthy controls (i.e., no clinical diagnosis of neurological disease or pathological evidence of α-synucleinopathy), who had been prospectively collected by our group. The NOR samples were stored at our in-house brain bank. The Spanish cohort (ESP, n = 62) comprised 51 individuals with iPD and 11 neurologically healthy controls from the Neurological Tissue Bank, Hospital Clínic – IDIBAPS Biobank, Barcelona, Spain. For the snRNA analyses, two additional controls were added from the Medical Research Council London Neurodegenerative Disease Brain Bank, King's college London, Great Britain, and one additional control from the Netherlands Brain Bank, Amsterdam, The Netherlands.

Analysis of whole-exome sequencing data from the NOR cohort[55] and RNA-sequencing (RNA-seq) data from both cohorts (described below), revealed no pathogenic mutations in nuclear genes linked to monogenic PD or mitochondrial diseases, thus ruling out monogenic causes. Prospectively collected systematic clinical and environmental exposure data were available for the iPD cases of the NOR cohort, as previously described[54]. In the NOR cohort, probable PDD was diagnosed according to the Movement Disorder Society Level 1 criteria[8]. Cognitive deficiency severe enough to impair daily life was defined based on the Unified Parkinson's Disease Rating Scale (UPDRS) score 1-1[56]. Impairment in cognitive domains was determined based on tests of attention ("serial 7's" test), visuo-constructive ability (correct drawing of the pentagons), and memory impairment (3-word recall),

from the Mini-Mental State Examination (MMSE)[57]. Appropriate tests on executive function were not available, thus the diagnosis of PDD in the NOR cohort was based on data from three, rather than four cognitive domains, as required by the MDS Level 1 criteria[8]. Two individuals had impairment in only one cognitive domain in addition to fulfilling the remaining criteria for PDD. These two individuals may thus have been potentially misclassified and as non-PDD cases. Individuals were classified by motor phenotype as TD or AR according to the Schiess classification[9], and TD or PIGD according to the Jankovic classification[10]. Detailed retrospective clinical data were available on the controls of the NOR cohort from the hospital records. Retrospective clinical data on disease duration, dementia (yes/no) and PD phenotype (TD/AR/mixed) were available from the hospital records of individuals of the ESP cohort.

There was no significant difference between controls and iPD subjects in PMI or assigned sex distribution in either of the cohorts, or in the combined cohort. Both sexes were included in the study as available from our cohort. Age of death was similar between controls and iPD subjects in the NOR cohort, while the control subjects were older in the ESP cohort and in the combined cohort. Subsets of samples used in downstream analyses were matched for age of death, sex and PMI-range, with the exception of the transcriptomic analyses, where the controls were older as reflected by the age distribution in the ESP and combined cohorts. The demographic data of the cohorts and descriptive statistics of each cohort included in the IHC experiments are shown in Table 1.

Brains were collected at autopsy and processed according to either in-house[27] or Neurological Tissue Bank protocol[58]. For the neuropathological evaluation, Lewy and tau pathology were staged according to the Braak criteria[59–62], and β-amyloid phases were evaluated according to the Thal criteria[62,63]. All cases fulfilled the essential pathological criteria for PD: loss of the dopaminergic neurons of the SNpc in the presence of LP[64]. Subject demographics, pathology scores and clinical data are summarized in Supplementary Data 1. Experimental allocation of subjects is summarized in Supplementary Data 2.

### Immunohistochemistry
Immunohistochemistry (IHC) for mitochondrial markers was carried out on formalin-fixed, paraffin-embedded brain tissue from individuals of both the NOR and ESP cohorts. Serial 3 μm thick sections were stained with primary antibodies against the following epitopes; NDUFS4 (ab137064, polyclonal, dilution 1:1500, Abcam), NDUFB8 (ab110242, monoclonal, isotype IgG1, dilution 1:300, Abcam, validated in frozen IHC and tested in human samples), NDUFA9 (ab14713, monoclonal, isotype IgG1, dilution 1:750, Abcam), NDUFS1 (ab169540, dilution 1:300, Abcam), SDHA (ab14715, monoclonal, isotype IgG1, dilution 1:10,000, Abcam), UQCRC2 (ab14745, monoclonal, isotype IgG1, dilution 1:10,000, Abcam), MTCOI (459600, monoclonal, isotype IgG2a, dilution 1:15,000, Thermo Fisher Scientific, validated by Cell treatment), ATP5A (ab14748, dilution 1:20,000, Abcam), and VDAC1 (ab14734, monoclonal, isotype IgG2b, dilution 1:20,000, Abcam). Antibodies against NDUFS4, NDUFA9, NDUFS1, SDHA, UQCRC2, ATP5a and VDAC1 were all validated in IHC in paraffin embedded tissues and tested in human samples according to the supplier (Abcam).

Sections were stained in the intelliPATH FLX Automated Slide Stainer (Biocare). Deparaffinization and antigen retrieval was performed in the low pH EnVision FLEX Target Retrieval Solution at 98 °C for 24 min in the DAKO PT link from Agilent. Antibodies were diluted in DaVinci Green Diluent (Biocare, PD900) and incubated for 1 hour at room temperature. MACH4 Universal HRP-polymer (Biocare, M4U534) and DAB chromogen kit (Biocare DB801) were used for visualization. Tacha's Automated Hematoxylin (Biocare, NM-HEM) was used for visualization of nuclei. Sections were washed in the TBS Automation Wash buffer from Biocare (TWB945M). Experimental allocation is given in Supplementary Data 2.

## Image analysis

All sections were scanned using the NanoZoomer XR (Hamamatsu). Visual analysis was performed on all sections, and neurons with clearly visible nucleus and nucleolus were classified as either negative or positive (Fig. 1a). After classification was performed, the total percentage of positive neurons was calculated for each individual and area. From the prefrontal cortex (PFC) in the NOR cohort, three regions of 3.5 mm² spanning cortical layers 2–6 were chosen at a low magnification (1.5×), before analyses were carried out in the selected area at 40× magnification. Area selection from one region spanning 3.5 mm² was performed in a similar manner in the PFC from the ESP cohort, and in the occipital and temporal cortex, the cingulate gyrus, subiculum, putamen and the amygdala. All neurons per section were analyzed in the CA-areas of the hippocampus (CA4-1), globus pallidus, dentate nucleus, inferior olivary nucleus, and Purkinje cell layer of the cerebellar cortex. In the SNpc, all neuromelanin containing neurons were analyzed per section. SNpc sections with fewer than 25 identifiable neuromelanin containing neurons were excluded from downstream analyses of neuronal CI status because data resulting from these sections would entail high levels of statistical uncertainty, at the range of the expected group differences (i.e., any misclassified neuron as CI positive or negative would result in at least 4% (1/25) difference in the estimated proportion of positive neurons.) An uncertainty greater or equal to 4% was deemed unacceptable since it is close to the difference between the medians of the groups being compared in our analyses. Independent evaluation of the PFC from the NOR cohort was performed by two observers (IHF and CT). All analyses were carried out at 40× magnification. Visual analysis was carried out using the NDP.view2plus version(v)2.7.25 software (Hamamatsu).

## Single neuron laser-microdissection studies

Qualitative and quantitative mtDNA studies were performed in single laser-microdissected neurons from 20 μm thick sections of fresh-frozen tissue from the PFC, which had been stained with 0.25% cresyl violet in ddH2O as described[28]. Neurons were recognized based on their morphology. Laser microdissection was performed on a LMD7 microscope (Leica microsystems). A total of 162 cells (8–16 from each individual) were cut, of which 55 were from controls ($n = 7$) and 107 were from individuals with iPD ($n = 14$). Experimental allocation is shown in Supplementary Data 2. Single neurons were collected in the cap of 0.2 ml PCR tubes and lysed at 56 °C overnight in 15 μL lysis buffer (50 mM Tris pH 8.0, 0.5% Tween 20, 200 μg/mL proteinase K). Following lysis, cells were centrifuged at 2250 g for 10 min before proteinase K was deactivated at 95 °C for 10 min, and the samples were centrifuged again. Total mtDNA copy number and the fraction of major arc deletion were determined using a duplex real-time PCR assay which targeted the mitochondrial genes *MTND1* and *MTND4* as described[28]. Amplification was performed on a Step One Plus Real-Time PCR System (Applied Biosystems by Thermo Fisher Scientific) using TaqMan Fast Advanced Master Mix (Thermo Fisher Scientific). Thermal cycling consisted of one cycle at 95 °C for 20 s, and 40 cycles at 95 °C for 1 s and 60 °C for 20 s. The following primers and probes were used: MTND1: forward primer: 5'CCCTAAAACCCGCCACATCT-3', reverse primer: 5'-GAGCGATGGTGAGAGCTAAGGT-3', TaqMan MGB Probe: 5'-FAM-CCATCACCCTCTACATCACCGCCC-3'. MTND4: forward primer: 5'-CCATTCTCCTCCTATCCCTCAAC-3', reverse primer: 5'-CACAATCTGATGTTTTGGTTAAACTATATTT-3', TaqMan MGB probe: 5'-VIC-CCGACATCATTACCGGGTTTTCCTCTTG-3'. For absolute quantification, target amplification was compared with a standard curve made form a serial dilution containing equal amounts of PCR-generated and purified full length *MTND1* and *MTND4* template ($10^2$-$10^6$). The following primers were used for the standard curve; *MTND1*: forward primer: 5´-CAGCCGCTATTAAAGGTTCG- 3', reverse primer: 5´- AGAGTGCGTCATATGTTGTTC-3´; *MTND4*: forward primer: 5´-TCCTTGTACTATCCCTATGAG-3´, reverse primer: 5´- GTGGCTCAGT

GTCAGTTCG-3´. All primers and probes were custom made and purchased from ThermoFisher Scientific.

## RNA sequencing

RNA-seq was performed on frozen PFC tissue samples from 98 individuals from both cohorts (Supplementary Data 2). Total RNA was extracted from tissue homogenates using RNeasy plus mini kit (Qiagen) with on-column DNase treatment according to manufacturer's protocol. Final elution was made in 65 μl of ddH2O. The concentration and integrity of the total RNA were estimated by Ribogreen assay (Thermo Fisher Scientific), and Fragment Analyzer (Advanced Analytical), respectively, and 500 ng of total RNA was used for library prep using the TruSeq™ Stranded Total RNA With Ribo-Zero™ Plus rRNA Depletion protocol (Illumina). Library quantity was assessed by Picogreen Assay (Thermo Fisher Scientific), and the library quality was estimated by utilizing a DNA High Sense chip on a Caliper Gx (Perkin Elmer). Accurate quantification of the final libraries for sequencing applications was determined using the qPCR-based KAPA Biosystems Library Quantification kit (Kapa Biosystems, Inc.). Each library was diluted to a final concentration of 1.4 nM and pooled equimolar prior to clustering. Sequencing was performed on a NovaSeq 6000 (Illumina, Inc.), to yield 100 million, 100 base pair paired-end reads per sample. Sequencing was performed at HudsonAlpha Institute for Biotechnology, AL, USA. RNA integrity number (RIN) and DV200 index (percentage of fragments >200 nucleotides) were lower in the NOR cohort compared to the ESP cohort: (mean [min-max]), $NOR_{RIN}$: 4.3 [1.7–7.7], $ESP_{RIN}$: 6.9 [2.7–9.1]; $NOR_{DV200}$: 85.3 [71.8–93.2], $ESP_{DV200}$: 92.2 [80.1–96.2]. No significant difference in the RNA quality measures between individuals with iPD and controls was observed in either of the cohorts (linear model, $NOR_{RIN}$: $P = 0.26$, $NOR_{DV200}$: $P = 0.33$; $ESP_{RIN}$: $P = 0.11$, $ESP_{DV200}$: $P = 0.11$).

## RNA-seq analysis

Salmon v1.3.0 [58] was used to quantify the abundance at the transcript level with the fragment-level GC bias correction option (--gcBias) against GENCODE v35. Transcript-level quantification was collapsed onto gene-level quantification using the tximport R package v1.8.0[65] according to the gene definitions provided by the same Ensembl release. Non-canonical chromosomes and scaffolds, and genes encoded by the mitochondrial genome were excluded from the analysis. In addition, we removed genes accounting for >0.6% of the reads in more than half of the samples, resulting in the removal of 4 transcripts (*RN7SL2, RN7SL1, RN7SK, LINC00632*; Supplementary Fig. S9) as well as low-expressed genes (expression <5 reads in 80% of the samples). At the final step, we used the expression level of the sex-specific genes *XIST, KDM5D* and *RPS4Y1* to define the noise threshold for our data[32]. Relative cell abundance was estimated using cell-type specific marker gene profiles (MGPs), as previously described[32,66]. Difference in the cell type MGPs between the groups were assessed through linear regression adjusting for age, sex, PMI and cohort, using the "lm" function from the R "stats" package v4.1.0. Differential expression (DE) analysis between each of iPD clusters and controls was performed using the "DESeq2" R package v1.32, using the protein-coding genes to calculate the normalization factor, while adjusting for sex, age, PMI, cohort and the relevant MGPs. Functional enrichment analysis was performed based on the one-sided *P*-value as previously described[32,33], using the gene score resampling method implemented in ermineR, an R wrapper package for ermineJ[67]. Enrichment was calculated for the combined list of gene-sets of Gene Ontology (GO) database[68] and Kyoto Encyclopedia of Genes and Genomes (KEGG) pathways[69,70] v7.2 as well as An Inventory of Mammalian Mitochondrial Genes v3 (MitoCarta3.0)[71]. Only gene-sets of size ≥20 and ≤500 were included in the analysis, resulting in 5397 testable gene-sets. To visualize the enrichment results, pair-wise similarity between significant gene-sets (excluding gene-sets from the MitoCarta database) was first assessed

using Jaccard Index. Next, the obtained similarity matrix was used to cluster the gene-sets using the "hclust" function from the "stats" R package v4.1.2, with the "ward.D2" method. The phylo tree was then created using the "ape" R package v5.5. The count matrix and the code necessary to reproduce these results are available through github repository: https://github.com/gsnido/pd_complex-i_stratification. All the information regarding the version of the software and the packages used for the analysis are provided in the SessionInfo.Rds file.

## Single nucleotide variant discovery

RNA-seq reads were aligned against the hg38 reference genome using Hisat2 v2.2.1[72]. Genome Analysis Toolkit (GATK) v4.1.9.0[73] was used to mark duplicate reads and split the CIGAR signatures using the *MarkDuplicates* and *SplitNCigarReads* tools, respectively. Base quality score recalibration was then performed using the *BaseRecalibrator* and *ApplyBQRS* tools to generate the final alignment files. For each alignment, *HaplotypeCaller* was used in GVCF mode (option *-ERC GVCF*) with optional arguments *-dont-use-soft-clipped-bases* and *--max-alternate-alleles 4*. Joint genotyping was carried out for all resulting sample gVCF files using the *GenomicsDBImport* and *GenotypeGVCFs* tools. The final analysis-ready multisample VCF consisted only of single-nucleotide variants filtered using *VariantFiltration* with the conditions QD < 5; DP < 10; FS > 60; MQ < 40. Samples were classified into the major mitochondrial haplogroups using HaploGrep2[74] based on the Phylotree Build 17[75].

Genotypes with <10 reads were removed, and the genetic data was recoded into binary PLINK format using VCFTools v0.1.15[76]. PLINK v1.90 was used for individual and variant-level quality control (QC)[77]. Only autosomes were considered, and variants within the sex chromosomes and mtDNA were removed. Individuals were excluded if their genomic data showed a missing rate >20%, abnormal heterozygosity (±3 standard deviations) or cryptic relatedness (identity by descent (IBD) > 0.2). Population stratification was studied using multidimensional scaling (MDS) against the HapMap populations[78].

Single nucleotide variants (SNVs) were excluded if the missing genotype rate was >20%, the call rates between PD and controls differed significantly ($P < 0.02$) or they displayed departure from the Hardy-Weinberg equilibrium ($P < 1.0 \times 10^{-5}$). Monomorphic variants were removed, and the dataset contained no multiallelic variants after QC. Principal component (PC) analysis was performed using Eigensoft[79,80].

## Single-nuclei isolation

Nuclei isolation procedure was performed as described in[81] with minor modifications. For each sample, 50–85 mg of frozen human prefrontal cortex were homogenized. All procedures were performed on ice. Tissue was dissociated in 2 mL of homogenization buffer (320 mM sucrose, 5 mM CaCl2, 3 mM Mg(CH3COO)2, 10 mM Tris HCl [pH 7.8], 0.1 mM EDTA [pH 8.0], 0.1% IGEPAL CA-630, 1 mM DTT, and 0.4 U/μL RNase Inhibitor (ThermoFisher Scientific) inside a 7 mL KIMBLE Dounce tissue grinder set (Merck), using 10 strokes with loose pestle followed by 10 strokes with tight pestle. Homogenized tissue was filtered through a 30 μm cell strainer and centrifuged at 1000 × *g* for 8 min at 4 °C. Supernatant was discarded, and pellet was resuspended in 450 μL of 2% BSA (in 1× PBS) containing 0.12 U/μL RNase Inhibitor. Cleanup of the nuclei suspension was achieved with density gradient centrifugation. Equal volumes (450 μL) of nuclei suspension and working solution (50% OptiPrep density gradient medium (Merck), 5 mM CaCl2, 3 mM Mg(CH3COO)2, 10 mM Tris HCl [pH 7.8], 0.1 mM EDTA [pH 8.0], and 1 mM DTT) were mixed. The density gradient was prepared by carefully layering 750 μL of 30% OptiPrep Solution (134 mM sucrose, 5 mM CaCl2, 3 mM Mg(CH3COO)2, 10 mM Tris HCl [pH 7.8], 0.1 mM EDTA [pH 8.0], 1 mM DTT, 0.04% IGEPAL CA-630, and 0.17 U/μL RNase Inhibitor) on top of 300 μL of 40% OptiPrep Solution (96 mM sucrose, 5 mM CaCl2, 3 mM Mg(CH3COO)2, 10 mM Tris HCl

[pH 7.8], 0.1 mM EDTA [pH 8.0], 1 mM DTT, 0.03% IGEPAL CA-630, and 0.12 U/μL RNase Inhibitor) inside a 2 mL Sorenson Dolphin micro-centrifuge tube (Merck). The mixture containing the nuclei (800 μL) was slowly pipetted onto the top of the OptiPrep density gradient and samples were centrifuged at 10,000 × *g* for 5 min at 4 °C using a fixed angle rotor (FA-45-24-11-Kit). Nuclei were recovered by withdrawing a volume of 200 μL from the 30–40% interface of the OptiPrep density gradient and transferring to a new 2 mL Sorenson Dolphin micro-centrifuge tube (Merck). Nuclei were washed with 2% BSA (in 1× PBS) containing 0.12 U/μL RNase Inhibitor and then pelleted by centrifugation at 300 × *g* for 3 min at 4 °C using a swing-bucket rotor (S-24-11-AT) for a total of three times. The nuclear pellet was resuspended in a minimum volume of 100 μL of 2% BSA (in 1× PBS) containing 0.12 U/μL RNase Inhibitor and carefully mixed by pipetting 20 times on ice. The nuclei suspension was filtered through a 40 μm FlowMe cell strainer (Merck) to remove any remaining nuclei aggregates. Nuclei were stained with Trypan blue and counted manually on a hemocytometer.

10,000 nuclei per sample were targeted for droplet-based snRNA sequencing. cDNA synthesis and libraries were prepared using the Chromium Next GEM Single Cell 3′ Reagent Kit v3.1 Dual Index (10× Genomics) according to the manufacturer's protocol. Libraries from each sample were pooled and run using paired-end (300 million read pairs per sample) sequencing on the NovaSeq 6000 platform (Illumina) at Novogene (Cambridge).

## snRNA-seq analyses

Raw FASTQ files were aligned to the human genome (GRCh38) using CellRanger v.6.1.2 (10× Genomics) accounting also for intronic reads to include all nuclear transcripts (include-introns option) and excluding secondary mappings (nosecondary option). In order to estimate intronic:exonic ratio, an additional alignment was carried out only accounting for reads mapped to exonic reads. Version 35 of the ENCODE transcriptome assembly was used (Ensembl 101).

To minimize the confounding effect of non-nuclear ambient RNA and discard empty droplets, we employed CellBender[82] on the raw data prior to importing the counts into R. Percentage of mitochondrially-encoded genes and ribosomal genes were then calculated for each droplet using the Seurat R Package v4.3.0.1 (*PercentageFeatureSet*)[83]. Each biological sample was independently normalized, and automatic feature selection was carried out using Seurat (*NormalizeData*, *FindVariableFeatures*). In order to collectively analyze the datasets, samples were integrated using the standard Seurat integration strategy. Briefly, features (genes) were automatically selected using *SelectIntegrationFeatures* to scale each sample and obtain a principal component analysis (PCA) of gene expression. Using the first 50 principal components, samples were integrated using reciprocal PCA with automatically detected anchors (*FindIntegrationAnchors*) and using controls to build the reference (*IntegrateData*). Clusters were first identified using the standard Seurat analysis with excess barcodes (i.e., including twice as many barcodes in each sample to ensure the inclusion of a large population of empty droplets). On each sample the largest cluster always corresponded to the cluster with the largest proportion of empty droplets (as defined by DropletUtils[84]). Other smaller ambient clusters were removed together with the main empty droplet cluster for downstream analyses. Resulting non-empty droplets were then further screened and only droplets with at least 500 transcripts, at least 1000 UMIs, and below 3% mitochondrially-encoded transcripts were analyzed. Transcripts detected in fewer than 10 cells were also filtered out. Doublets were assessed independently for each sample using the DoubletFinder R package[85] assuming 7% doublet rate. Standard clustering using Seurat with 0.5 resolution resulted in a total of 28 clusters (Fig. 7a) which could be assigned to the main cortical cell types based on known markers (SATB2 and SLC17A7 for excitatory neurons, GAD1 and

SLC6A1 for inhibitory neurons, MOG and MOBP for oligodendroglia, GFAP and AQP4 for astrocytes, CLDN5 and SLC2A1 for endothelial cells, OLIG1 and VCAN for oligodendrocyte precursor cells, and CSF1R and CSF3R for microglia[36], Fig. 7b). Finally, to increase the within-cluster cell type homogeneity and minimize the number of doublets, we used two alternative approaches. First, the Euclidean distance in PCA space was calculated between each single cell and the average of each cluster using the first 5 PCs. Barcodes for which the shortest distance did not correspond to their cluster membership were removed from the analyses. We then calculated the first PC of the cell type specific markers for each barcode (marker gene profiles, MGP[66]). MGPs correspond to the first PC of the cell type specific markers. In this way, we obtained for each droplet a value for each major cell type (i.e., neuron, oligodendrocyte, endothelial cell, microglia, astrocyte, OPC) scaled across all droplets. We then calculated a score for each droplet, corresponding to the MGP for the annotated cell type divided by the sum of the other MGPs. Droplets with poor scores (i.e., below 0.4) were filtered out. The filtering procedure resulted in a total of 117,105 nuclei. The cluster labels were used to compare cell type composition of the diseased and control groups using ANOVA.

To assess the statistical significance of the difference in the number of captured nuclei between groups (nCI-PD, CI-PD and Ctrl) we used an ANOVA test at $\alpha = 0.05$. Differences between groups in the number of genes per nucleus were assessed for each cell type cluster with linear mixed-effects models (LMM) with disease status as fixed effect and individual as random effect. $P$-values for each LMM were estimated using the Satterthwaite's method. The resulting $p$-values were considered significant above FDR of 5%. The same statistical approach was used to assess differences in the number of UMI counts between disease groups.

Differential expression within clusters between diseased groups and controls was carried out using the MAST R package[86] using sex, post-mortem interval and age as fixed effects, and individual as random effect[87] (Fig. 7c). For each cluster, only autosomal, protein-coding genes that were expressed were considered (i.e., with counts >0 in 25% of the barcodes). The same methodology was used to compare between the CI-PD and nCI-PD subtypes. Overrepresentation analysis was carried out using the fora function from the fgsea R package[88] independently for each cell type cluster. The background ("universe") for each overrepresentation test consisted of the set of expressed genes for each specific cell type cluster.

UMI counts and code necessary to reproduce these results is available through a github repository: https://github.com/gsnido/pd_complex-i_stratification.

### Statistical analyses and reproducibility of the demographic, clinical, IHC and mtDNA data

Subject demographics, clinical data, and pathology scores are given in Supplementary Data 1. Experimental allocation is given in Supplementary Data 2. The sample size was not predetermined. No data were excluded from the analyses. Apart from the IHC staining of the tissues from the ESP cohort, the experiments were not randomized. Observer 1 was blinded to allocation during IHC staining and outcome assessments of the CI positivity in the ESP cohort. Observer 2 in the IHC analyses of CI positive staining was blinded to group specificity. Investigators were blinded during the preprocessing of the RNA-seq data. Apart from this, the investigators were not blinded to allocation during experiments and outcome assessments. The visual analysis of the IHC stainings for CI was performed by two independent observers (IHC and CT). IHC findings from the NOR cohort were replicated in the ESP cohort. The samples from single cell qPCR were run in triplicates on the same 96 well plate.

Data were tested for normality and equality of variances, using the Shapiro-Wilk test and Levene's test. Clustering was performed in R

(v4.1.2), using k-means with three centers, 20 random starting partitions, 100 iterations, and the Hartigan-Wong algorithm. Between group-comparisons were consequently performed using either independent student's t-test or Mann–Whitney U-test, depending on data normality, and descriptive data are given either as mean (standard deviation) or median [25th-percentile-75th-percentile], respectively. Comparisons of grouped data were performed using either Chi-Square ($X^2$) or Fisher's exact test depending on the group size of each variable. Data is given as $X^2$ and/or odds ratio (95% confidence interval). Correlation analyses were performed using Spearman's rank correlation due to non-normality of the data, Spearman's rank correlation coefficient is given as $\rho(n - 2)$. Between-group (Ctrl, CI-PD and nCI-PD) comparison of neuronal mtDNA deletion proportion and copy number was performed by one-way ANOVA on the individual median values. Due to the unequal sample sizes, Welch's test for unequal variance was used. Eta squared ($\eta^2$) was used to determine the effect size of any significant difference. The Games-Howell post hoc test was used to assess the statistical significance of pairwise differences between groups, due to their unequal sample sizes and variance. Hierarchical clustering analysis was performed in R v4.1.2 with the "ward.D2" method. Correlation analyses and between-group comparisons were done in SPSS v28.0.0. Plots were created in R v4.1.2 or GraphPad Prism v9.3.0. A $P$-value < 0.05 was considered statistically significant. All tests are 2-sided unless else is noted in the text. Statistical tests performed in the context of omics data analyses are described in the corresponding methods sections.

### Reporting summary

Further information on research design is available in the Nature Portfolio Reporting Summary linked to this article.

## Data availability

The IHC and mtDNA qPCR data generated in this study are provided in the Supplementary Information/Source Data file. The raw and filtered counts from the transcriptomic data are available at Synapse.org: (https://doi.org/10.7303/syn53502566.1)[89]. The raw transcriptomic data of the bulk tissue samples (accession number: EGAD50000000432) and the single nuclei samples (accession number: EGAD50000000433) are available in the Federated European Genome-phenome Archive (FEGA) Norway. Source data are provided with this paper.

## Code availability

The code required to reproduce the results of the transcriptomic analyses are available in github.com: https://github.com/gsnido/pd_complex-i_stratification (https://doi.org/10.5281/zenodo.10715023)[90].

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

## Acknowledgements

We are deeply grateful to the study participants and their families involved in the study for their unique contribution. Furthermore, we would like to thank the entire Park West study consortium for their diligent efforts characterizing and following the Park West cohort; Dr. Yamila Torres Cleuren for the inspirational feedback and critical review of the manuscript; the Neurological Tissue Bank of the IDIBAPS-Hospital Clinic Biobank for providing data and samples; Dr. Brian L Edlow and colleagues for allowing us to use images from their 7 Tesla MRI atlas in Fig. 3; Dr. Romain Guitton, Dr. Kristoffer Haugarvoll and Janani Sundaresan for helping dissect and prepare the brain tissue for RNA-seq; Gry Hilde Nilsen for technical support. This work is supported by grants from The Research Council of Norway (288164; CT), Bergen Research Foundation (BFS2017REK05; CT), The KG Jebsen Foundation (SKGJ-MED-023; CT), and the Western Norway Regional Health Authority (F-10229-D11661; IHF).

## Author contributions

I.H.F.: participated in the study conception and design, designed and performed the IHC and qPCR experiments, assessed the neuropathology of the NOR cohort, analyzed the IHC, mtDNA, clinical and pathology data, and drafted the manuscript. L.T.: participated in the study design,

analyzed and interpreted the transcriptomics data, participated in, and advised on statistical approaches, and drafted parts of the manuscript. EFV: critical revision of the manuscript. D.A.S. and O.S.: performed part of the IHC stainings. M.G. and S.M.: nuclei isolation from brain tissue and library preparation for snRNA-seq. N.L.: contributed to the preparation of Fig. 2, advised on the statistical approach, and provided critical input for the manuscript. G.A. and O.B.T.: contributed biological material and clinical data and provided critical input to the manuscript. A.P.-S., C.P., Y.C.: contributed clinical data and input to the manuscript. L.M.P.: contributed biological material, pathological data and critical input to the manuscript. C.D.: contributed to the qPCR analyses and provided critical input for the manuscript. G.S.N.: participated in the study design, performed the prepossessing of all transcriptomic and genetic data, performed the mtDNA variant analyses, participated in the interpretation of the transcriptomic and genetic data, analyzed the single nuclei transcriptomics data and drafted parts of the manuscript. C.T.: conceived, designed and directed the study, contributed to data analyses and interpretation, drafted the manuscript, and acquired funding for the study. All authors have read and approved the manuscript.

## Funding

## Competing interests
The authors declare no competing interests.
