## [Peer Review File · Nature Communications]

Mitochondrial complex I deficiency stratifies idiopathic Parkinson's diseaseReviewers' comments:

Reviewer #1 (Remarks to the Author):

The authors have sought to define subgroups of Parkinson's disease (PD) patients according to drivers of certain pathogenetic pathways. The notion that PD is a heterogeneous disease in etiological terms is not new; several different causative (monogenic) genes and a number of risk genes have been identified. Many of these genetic associations map to certain pathways, the mitochondrial and lysosomal in particular. The authors might approach their introduction in this way, as this provides a sound basis to seeking to categorize PD etiopathogenesis.

In this study, mitochondrial dysfunction and Complex I deficiency in particular was defined by the staining intensity of a single nuclear-encoded subunit, NDUFS4. Mitochondrial content was measured by intensity of VDAC1, an outer mitochondrial membrane protein. Samples were studied from the SNc and the prefrontal cortex (PFC). The authors should provide more detail on the precise anatomical location of their samples e.g. SNc-pars compacta? A10?

It is surprising that at the group level, the SNc exhibited pronounced CI deficiency in both iPD and control subjects. The SNc is regarded as a site of high oxidative phosphorylation capacity. This would decline with poor tissue preservation or agonal state, despite the authors' assertion to the contrary in their study. It is notable that the larger number of controls came from the Norwegian bank with a long post-mortem delay of 43h, compared to the smaller Spanish group of 7.5h, was there a statistical difference between these two groups? Similarly, the Norwegian PD group delay was 30h versus the Spanish delay of 8h. was this statistically different? The two brain sources seem to be very imbalanced. The NDUFS4 staining was lower than control in the Norwegian but not the Spanish brains.

The reasons for the selection of PFC over SNc for analyses is not altogether convincing. As the authors state, the SNc is a site of major pathology and a driver for clinical features. The PFC probably plays a peripheral role in comparison. The authors should at least provide data on both areas in their further comparisons.

The idea that Complex I deficiency is not relevant to all iPD cases is not of course new. The authors do not reference the original discoveries of Complex I deficient activity in the PD SNc by the London group, this should be rectified. This group also subsequently analyzed respiratory chain activity in PD platelets and showed that around 25% had complex I deficiency, an observation that would support the current study.

Analysis of additional respiratory chain proteins was limited to only two to four brains, and anatomical distribution of Complex I to 15 brains. It is difficult to make clear judgements on the relevance of the findings with such low numbers. Similarly, the mtDNA qPCR data were based on only 6 NDUFS4 deficient, 9 normal staining, and 7 controls, with only 8-16 neurons sampled per brain. It is a shame that the immunostaining did not include a mtDNA-encoded subunit as well for these analyses, perhaps one included in the 'common' deletion site.

A major limitation of this study is that the primary analyses and segregation into Complex I and non-Complex I related PD was based on a single antibody immunostain. There was no attempt to supplement this with actual activity data, which presumably could have been done with the fresh-frozen samples from the same brains. If not available, then the study should have been extended to those brains from

which both fresh-frozen and fixed samples existed.

Reviewer #2 (Remarks to the Author):

Flonas and colleagues evaluated the existence of respiratory complex I deficiency in different brain regions from two idiopathic Parkinson cohorts. They propose here the existence of two PD-CI subtypes: (1) One with high CI deficient, (less frequent) affecting different brain regions (2) another one with CI deficient limited to SN. They found that the existence of both subtypes is not due to differences in the disease course. They show that both subtypes are associated with specific clinical subtypes, different lifestyle risk effects, mtDNA deletions and different transcriptional signatures. The paper is written concisely and clearly.

However, I have some significant concerns. Firstly these subtypes have been established artificially and have not been appropriately validated. Furthermore, even true it is hard to see this study's clinical impact: these observations are based on a few IC markers on postmortem brain tissue at an advanced disease stage, in the small cohort, with relatively sparse clinical information compared with other deeply phenotyped cohorts (e.g. PPMI)

Major

(1) I am concerned about how both subtypes have been statistically established and validated: (i) The authors should determine two clusters with only one cohort and then validate the existence in the second cohort (ii) With the previous studies published:

They should demonstrate statistically how the existence of these subtypes can explain that the CI deficiency has not been observed in 50% of the previously published studies

Flonas et al should re-analysed all the previous CI studies published to validate the evidence of the existence of both subtypes. Are the nonCI/CI associated with the same clinical subtype/genetic subtype or transcriptomic signatures?

(2) The CI subtype is likely associated with mitochondrial deletions. It is regrettable that the authors don't take advantage of single technologies to characterise mtDNA at this single level PMID: 35181467

(3) Both cohorts are not clinically characterised enough (unlike PPMI) to decide which clinical feature is the most relevant for each subtype.

(4) In many cases, It is not clear for different comparisons how the authors adjusted the p-value for the multiple testing

(5) We can see there are more males than females in nCI than in CI, which can explain the difference in alcohol consumption and cigarettes

Minor comments

- (1) What does mean neurologically healthy controls – individuals affected by other neurological conditions
- (2) To avoid using the abbreviations in the title e.g CI (abbreviation also used for Confidence Interval
- (3) The comparison can not be only qualitative. Please mention in the text for each comparison (1) the directionality (2) the significance (3) the method of adjustment of p-value Bonferroni vs FDR

Reviewer #3 (Remarks to the Author):

This is an interesting manuscript which provides new information around complex I deficiency in iPD. This is an area where research is challenging because of the use of post-mortem tissue and the heterogeneity between patients and even between controls. Use of human tissue is essential since animal models do not help with these age related changes.

I have the following comments:

1. The authors have used two well characterised cohorts of patients for their studies validating their results which is I think optimal for these sort of studies. The fact that they see clear evidence of complex I deficiency in both cohorts is important
2. I am somewhat concerned about their use of only immunohistochemistry for defining complex I deficiency, and indeed evidence of changes in other complexes. I know the authors have used this technique for other studies but there are more quantitative ways of measuring protein abundance in individual cells using fluorescent antibodies and image analysis software. Use of these techniques would allow a more accurate assessment of complex I (or other deficiencies). At least I think they have to better justify their approach or preferably it would be nice to see a limited number of cases assessed using more quantitative techniques
3. Their analysis of single cell mtDNA is interesting especially using the PFC – the difficulty is that they cannot directly look at those neurons which are complex I deficient. They are correct in that we do not know whether the deletion causes the deficiency or the complex I deficiency causes the deletion. This raises the interesting question, which I do not think is tackled in the manuscript, is what is causing the complex I deficiency. I know not a central point of the manuscript but interesting to see the authors speculations
4. Another point worth the authors discussing is around the mechanism by which the complex I deficiency could be affecting the phenotype of iPD? Is the deficiency seen in different areas of the brain responsible?
5. The authors make a very good point about the heterogeneity of complex I deficiency is important if considering clinical trials and their results. The authors suggest that agents might need to be targeted to those where there is the most evidence of complex I deficiency which is true. However it is important that the authors recognise that at present there is no effective treatment for complex I deficiency.

Reviewer #1 (Remarks to the Author):

Comment-1:

The authors have sought to define subgroups of Parkinson's disease (PD) patients according to drivers of certain pathogenetic pathways. The notion that PD is a heterogeneous disease in etiological terms is not new; several different causative (monogenic) genes and a number of risk genes have been identified. Many of these genetic associations map to certain pathways, the mitochondrial and lysosomal in particular. The authors might approach their introduction in this way, as this provides a sound basis to seeking to categorize PD etiopathogenesis.

Response-1:

We thank the reviewer for this suggestion. Since the first submission, we have partly restructured and compacted our introduction. We do mention the major mechanisms associated with iPD: mitochondria, proteostasis (lysosome/proteasome), neuroinflammation, and subsequently focus on mitochondrial dysfunction, briefly summarizing the evidence and highlighting the fact that its pervasiveness is not established.

We would like to respectfully point out that we have not claimed the notion of PD being an etiologically heterogeneous disease is new. In fact, we cite other studies supporting this notion.

Comment-2:

In this study, mitochondrial dysfunction and Complex I deficiency in particular was defined by the staining intensity of a single nuclear-encoded subunit, NDUFS4. Mitochondrial content was measured by intensity of VDAC1, an outer mitochondrial membrane protein. Samples were studied from the SNc and the prefrontal cortex (PFC). The authors should provide more detail on the precise anatomical location of their samples e.g., SNc-pars compacta? A10?

Response-2:

We acknowledge that presenting results from a single complex I subunit is a weakness and have rectified this limitation. We have now tested antibodies against multiple CI subunits (NDUFS4, NDUFS1, NDUFA10, NDUFA9 and NDUF8), across the functional domains of the complex of PD patients and controls. All tested subunits exhibited a similar pattern and severity of neuronal CI deficiency, consistent with a quantitative reduction of the entire CI in affected cells (Supplementary Fig. S1). Subunit NDUFS4 was chosen for downstream analyses due to superior staining quality. The entire SN pars compacta was assessed. This is now specified in the text.

Comment-3:

It is surprising that at the group level, the SNc exhibited pronounced CI deficiency in both iPD and control subjects. The SNc is regarded as a site of high oxidative phosphorylation capacity. This would decline with poor tissue preservation or agonal state, despite the authors' assertion to the contrary in their study. It is notable that the larger number of controls came from the Norwegian bank with a long post-mortem delay of 43h, compared to the smaller Spanish group of 7.5h, was there a statistical difference between these two groups? Similarly, the Norwegian PD group delay was 30h versus the Spanish delay of 8h. was this statistically different? The two brain sources seem

to be very imbalanced. The NDUFS4 staining was lower than control in the Norwegian but not the Spanish brains.

Response-3:

We respectfully disagree that the finding of CI deficiency in the control SNc is surprising. The dopaminergic SNc is known to exhibit high levels of CI deficiency with healthy aging and this has been reported by multiple studies¹⁻³. It is true that PD has been shown to harbor even higher levels of CI deficiency in the SNc. However, the overlap with the neurologically healthy aged population is substantial. In contrast, the PFC exhibits CI deficiency with much higher disease-specificity and is, therefore, better suited for classification.

It is true that the Norwegian cohort had a higher PMI than the Spanish cohort, however this difference is not important since both stratification and downstream comparisons were made between PD and controls **within** each cohort. There was no significant difference between the PMI of cases and controls in either the Norwegian (Ctr: 49.0 ± 25.6 , PD: 40.4 ± 22.6 , $p = 0.26$) or the Spanish cohort (Ctr: 8.3 ± 3.1 , PD: 10.2 ± 5.1 , $p = 0.24$). This is illustrated below in **Figure 1**. Moreover, PMI was not associated with the severity of CI deficiency in either the SN ($p = 0.10$), or the PFC ($p = 0.31$).

Figure 1. Distribution of PMI between individuals with PD and controls in each of the Norwegian (NOR) and Spanish (ESP) cohorts. A: box plots showing the median and IQR. B: Histograms showing the distribution of PMI across individuals. There is no significant difference between PD and controls in either of the cohorts. Ctr: controls. PD: Parkinson's disease.

Comment-4:

The reasons for the selection of PFC over SNc for analyses is not altogether convincing. As the authors state, the SNc is a site of major pathology and a driver for clinical features. The PFC probably plays a peripheral role in comparison. The authors should at least provide data on both areas in their further comparisons.

Response-4:

The goal of the classification was to stratify into subgroups as reliably and objectively as possible, before subtype-specific disease mechanisms are explored. CI deficiency in the PFC performs better as a classifier of PD cases for several reasons:

1) Neuronal CI deficiency was more disease-specific in the PFC than in the SNc, which overlapped substantially with the demographically matched controls (see also response to Comment-3, above). The fact that the SNc exhibits high levels of CI deficiency and other markers of mitochondrial dysfunction (regardless of whether these are significantly higher in PD or not) also with healthy aging, complicates classification and makes it challenging to distinguish between disease-specific and age-related changes.

2) At the time of autopsy, the SNc typically exhibits ~80% loss of its dopaminergic neurons⁴. This severe degeneration greatly complicates the comparison between cases and controls, as the few surviving SNc neurons are likely to be resilient cells, introducing survival bias. This is readily observable in our data. Namely, we observed that iPD cases with severe neuronal loss in the SNc had few to no CI negative dopaminergic neurons (in fact, in SNc with fewer than 25 surviving neurons, all remaining neurons were CI positive), suggesting that CI neurons are more susceptible to degeneration and death than CI intact neurons (see Supplementary Table 3). Thus, the SNc is not suitable for classifying our samples, as it would be introducing significant bias to the analyses.

Comment-5:

The idea that Complex I deficiency is not relevant to all iPD cases is not of course new. The authors do not reference the original discoveries of Complex I deficient activity in the PD SNc by the London group, this should be rectified. This group also subsequently analyzed respiratory chain activity in PD platelets and showed that around 25% had complex I deficiency, an observation that would support the current study.

Response-5:

We fully agree with the reviewer that the original findings by the London group should be referenced. We have always done so in our papers, and this was a mistake on our part. We apologize for this omission and have rectified it.

We are aware that the notion that CI deficiency/mitochondrial dysfunction may not be pervasive in iPD has been the subject of discussions in the field. However, to the best of our knowledge, no robust evidence has been published to date showing this to be a fact.

Regarding the Reviewer's second point, we are excited to know about the finding of CI deficiency in the platelets of ~25% of PD patients. However, despite our best efforts, we were unable to find a published paper describing these results. If the Reviewer would kindly point to that publication, we will be happy to refer to it in our paper.

Comment-6:

Analysis of additional respiratory chain proteins was limited to only two to four brains, and anatomical distribution of Complex I to 15 brains. It is difficult to make clear judgements on the relevance of the findings with such low numbers. Similarly, the mtDNA qPCR data were based on only 6 NFUFS4 deficient, 9 normal staining, and 7 controls, with only 8-16 neurons sampled per brain. It is a shame that the immunostaining did not include a mtDNA-encoded subunit as well for these analyses, perhaps one included in the 'common' deletion site.

Response-6:

The analyses of the respiratory complexes II-V were performed in a total of 9 brains (i.e., 2-4 individuals per group). Given that these complexes have not shown evidence of significant deficiency in the PD brain in previous studies by us and others^{1,5}, it is highly unlikely that a higher number of cases would change our results. We, therefore, deemed this number sufficient for the purpose of this work.

The anatomical distribution of CI deficiency was assessed in 17 individuals and up to 15 brain regions per individual. We would like to respectfully point out that this is, in fact, a large number of samples for this type of comprehensive and cumbersome histopathological analyses. Most importantly, the number of individuals and samples analyzed is adequate for revealing the anatomical distribution of neuronal CI deficiency in CI-PD and nCI-PD. That said, we would be happy to increase the number of samples, if deemed absolutely necessary.

The single-cell mtDNA analyses were done on a total of 8-16 neurons per individual and a total of 22 individuals (6 x CI-PD, 9 x nCI-PD, and 7 x controls). This sample size is appropriate for this type of analysis and was indeed powered to detect a significant difference between the groups.

Thus, we do not believe that increasing the sample size will provide more clarity on this question. On the other hand, differentiating between CI-positive and CI-negative cells would add valuable new insight. However, this has so far proven to be technically challenging (see our response to Comment-3 by Reviewer-3).

Regarding the immunostaining, as mentioned above - in response to Comment-2, we have now extended our analyses and tested antibodies against five CI subunits (NDUFS4, NDUFS1, NDUFA10, NDUFA9 and NDUFB8), across the functional domains of the complex. All tested subunits exhibited a similar pattern and severity of neuronal CI deficiency, consistent with a quantitative reduction of **the entire CI** in affected cells (Supplementary Fig. S1). We did, in fact, test several antibodies against an mtDNA-encoded (ND1) subunit, but none of them provided successful staining. Nevertheless, we have demonstrated the lack/reduction of five CI subunits (including a core subunit) spanning the functional domains of the complex, which can only be interpreted as a loss of the entire complex. We, therefore, may assume that our immunostaining results are representative of all CI subunits.

Comment-7:

A major limitation of this study is that the primary analyses and segregation into Complex I and non-Complex I related PD was based on a single antibody immunostain. There was no attempt to supplement this with actual activity data, which presumably could have been done with the fresh-frozen samples from the same brains. If not available, then the study should have been extended to those brains from which both fresh-frozen and fixed samples existed.

Response-7:

Functional assessment of specific CI activity was not undertaken in this work, because it is currently not feasible to assess in individual cells. Measurement of CI activity in bulk tissue would not be sufficiently reliable, and may produce misleading results, because of: 1) low sensitivity, due to the mosaic distribution of the CI defect, and 2) low specificity in the sense that it cannot adequately distinguish between cell-specific reduction in CI activity or a decrease in neuronal and synaptic content in the sample. While specific complex activities can be normalized for mitochondrial content (via ratio to citrate synthase), there is substantial evidence that different cell types have different compositions and organizations of mitochondrial respiratory chain, including

distinct respiratory complex stoichiometries. In neurons, the proportion of complex I which is organized in supercomplex with complex III is much higher than in astrocytes, resulting in higher bioenergetic efficiency⁶⁻⁸. Furthermore, differences in the composition of the respiratory chain occur between different neuronal compartments. Neuronal synapses have been shown to be more dependent on complex I driven respiration, compared to neuronal somas⁹. Thus, it stands to reason that specific complex I activity measurements in bulk brain tissue will be influenced by the underlying cell-type and synaptic composition of the tissue being tested.

The PFC exhibits substantial synaptic loss in PD due to cholinergic and noradrenergic denervation¹⁰. Moreover, we have previously shown that the PFC of iPD exhibits markers of decreased neuronal content¹¹, and our current data suggests that these differences may be more pronounced in CI-PD. Therefore, respiratory complex activity measurements in bulk tissue from CI-PD and nCI-PD would be biased by differences in cell composition, with the results being difficult to interpret at the cell-specific level.

Having said the above, we agree with the Reviewer that the classification into CI-PD and nCI-PD should not rely on a single measure. The classification is supported by multiple other datasets, unrelated to the immunohistochemical staining:

1. CI-PD and nCI-PD show different mtDNA profiles: increased load of neuronal mtDNA deletions in CI-PD
2. CI-PD and nCI-PD show highly distinct transcriptomic profiles in bulk tissue.
3. CI-PD and nCI-PD show different clinical features, with a strong predilection for non-tremor dominant motor phenotypes in CI-PD.

Furthermore, in this updated version of the manuscript we have added new data further supporting the existence of the subtypes:

1. We have increased the robustness of the IHC by showing concordance between multiple CI subunits, consistent with a loss of the entire complex in affected cells.
2. To mitigate the problem of heterogeneous cell-composition in the bulk-tissue, we performed single-nucleus massively parallel RNA sequencing. This data shows a highly distinct cell type-specific gene expression profile between the CI-PD and nCI-PD groups.

In summary, we have provided evidence from three different disciplines (histological, molecular, clinical) and five distinct information layers and methodologies (immunohistochemical quantification, mtDNA, bulk-tissue transcriptomics, single-nucleus transcriptomics, phenotypical data), all of which supports the distinct nature of the CI-PD and nCI-PD subtypes.

In our opinion, taken together, this data makes a very strong case for the existence of the CI-PD and nCI-PD subtypes. As is the case with any novel findings, the proposed CI-PD and nCI-PD groups will have to be independently validated and further explored by further research conducted by us and other groups in our field. We do, however, believe that the current evidence makes a sufficiently robust case to warrant publication and sharing of these novel findings with our peers.

Reviewer #2 (Remarks to the Author):

Flones and colleagues evaluated the existence of respiratory complex I deficiency in different brain regions from two idiopathic Parkinson cohorts. They propose here the existence of two PD-CI subtypes: (1) One with high CI deficient, (less frequent) affecting different brain regions (2) another one with CI deficient limited to SN. They found that the existence of both subtypes is not due to differences in the disease course. They show that both subtypes are associated with specific clinical subtypes, different lifestyle risk effects, mtDNA deletions and different transcriptional signatures. The paper is written concisely and clearly. However, I have some significant concerns. Firstly these subtypes have been established artificially and have not been appropriately validated. Furthermore, even true it is hard to see this study's clinical impact: these observations are based on a few IC markers on postmortem brain tissue at an advanced disease stage, in the small cohort, with relatively sparse clinical information compared with other deeply phenotyped cohorts (e.g. PPMI)

Major Comments

Comment-1:

(1) I am concerned about how both subtypes have been statistically established and validated:
(i) The authors should determine two clusters with only one cohort and then validate the existence in the second cohort

Response-1i:

We thank the Reviewer for the comments. The Reviewer is referring to a classical discovery-replication approach. In our study we have taken an even more stringent approach, i.e., parallel independent discovery in both cohorts, with nearly identical results obtained from parallel independent analyses.

We first clustered each of the Norwegian and Spanish cohorts independently (using k-means). This revealed the same pattern in each of the cohorts independently: PD individuals showed a much larger spreading than the controls, with many vases clustering with the controls and ~25% clustering away from the controls, into two clusters exhibiting mild and severe CI deficiency, respectively.

Having independently established these findings in both the Norwegian and Spanish cohorts, we subsequently performed clustering on the merged cohort, as larger samples sizes generally increase the precision of clustering algorithms. Nevertheless, the cohorts were so consistent, that the merged clustering revealed the same results (i.e., nearly all individuals clustered as per their individual cohorts). The approach undertaken and results in each cohort individually, as well as in the merged cohort, are shown in Fig-1 in the manuscript.

In addition, we have now strengthened the quality of our data by including results on inter-rater variability (Fig1C).

Given the highly consistent results from analyzing each cohort separately and together, we are not sure how to further strengthen these results. We, respectfully, do not grasp the Reviewer's suggestion to determine two clusters with only one cohort and then validate the existence in the second cohort, because we do not see how the clusters could be validated in such a manner.

Determining a threshold from the first cohort and then apply it to the second is not helpful, as this would not be based on the intrinsic properties of the second cohort. Perhaps the reviewer refers to training and replication design, but our methodology does not involve learning algorithms. IN conclusion, we would argue that independent parallel discovery, as we have done, is the most stringent and robust way to replicate the clusters.

(ii) With the previous studies published: They should demonstrate statistically how the existence of these subtypes can explain that the CI deficiency has not been observed in 50% of the previously published studies

Flones et al should re-analysed all the previous CI studies published to validate the evidence of the existence of both subtypes. Are the nonCI/CI associated with the same clinical subtype/genetic subtype or transcriptomic signatures?

Response-1ii:

We thank the Reviewer for this suggestion. We have indeed made several attempts to reanalyze data from previous studies, ever since when we made the first observations at the beginning of our analyses. Unfortunately, a proper reanalysis/meta-analysis of the data from previously published studies is not possible, because the vast majority only report data at the group level and summary statistics (i.e., individual values are not given). Nevertheless, data from previous studies support the existence of the CI-PD and nCI-PD subtypes. Regardless of whether these studies show a significant difference between PD and controls at the group level, the PD group tends to overlap substantially with the control group, indicating that some, but not all individuals with PD harbor CI deficiency. These studies have been recently reviewed in Subramanian et al.⁵ and we are referring to them in our paper.

We are motivated to contact the authors of previous studies and attempt to collect individual data points. So far, the outcome has been disappointing since most of these studies are old and the data no longer exists. We remain interested in collecting any data that may still be available and are making efforts to that end. This is, however, outside the scope of the present study and, if fruitful, we may present these results in a follow-up paper.

As for ascertaining whether the CI-PD and nCI-PD subtypes are associated with the same clinical and transcriptomic signatures in previous studies, unfortunately this is not possible, since such data is not available. The Reviewer may find a comprehensive overview of previous studies in Subramanian et al.⁵

Comment-2

(2) The CI subtype is likely associated with mitochondrial deletions. It is regrettable that the authors don't take advantage of single technologies to characterise mtDNA at this single level PMID: 35181467

Response-2:

We are respectfully surprised by this comment, since we have, in fact, performed single-neuron analyses on a large number of neurons and individuals, with results supporting the CI-PD and nCI-PD clusters. This was present in the original submission and has now been further strengthened. Neuronal mtDNA copy number and deletion burden were assessed by qPCR in single neurons microdissected from individuals with CI-PD (n = 6), nCI-PD (n = 9) and controls (n = 7). Eight to sixteen single neurons were assessed per individual. There was a significant difference in neuronal mtDNA deletions between the groups (ANOVA, $P = 0.037$, $\eta^2 = 0.30$), with post-hoc analyses

showing that the CI-PD group had significantly higher levels of mtDNA deletions compared to the nCI-PD group ($P = 0.037$; Fig. 5b). This is described in the Results (section called “*CI-PD and nCI-PD show different neuronal mtDNA profiles*”) and shown in Figure-5 and Supplementary Table 6.

Comment-3

(3) Both cohorts are not clinically characterised enough (unlike PPMI) to decide which clinical feature is the most relevant for each subtype.

Response-3:

The Norwegian cohort has excellent clinical data, as these samples originate from the ParkWest cohort, a prospective, longitudinal and systematically characterized cohort, which has been described in detail¹². The Spanish cohort is from the Barcelona Brain Bank and, like most brain banks, lacks prospectively collected systematic clinical information.

While we agree with the Reviewer that, in an ideal world, we would have prospective and systematic clinical information on both cohorts, it is important to consider what is currently realistic. There are presently very few cohorts with both prospective systematic clinical data and brain collection in adequate numbers. Park West is one of those rare and extremely valuable cohorts, which has accumulated a relatively large number of brains, because it has been going for almost 20 years (since 2004). There is no doubt that PPMI has excellent clinical information, but we believe that the comparison made by the Reviewer is somewhat unfair, since the PPMI has not yet accumulated a sufficiently large brain collection and is would, therefore, not have been useful in the present study.

In our data, we see a strong association between CI-PD and akinetic rigid and/or PIGD phenotype and this is replicated in both cohorts. We believe that this is already important novel insight into the clinical correlates of the CI-PD subtype. That said, observations in much larger cohorts (several hundreds) will be necessary to identify the full clinical phenotype associated with CI-PD and nCI-PD. Such cohorts, combining prospective, systematic clinical data and brain collection from hundreds of individuals with PD and controls do not currently exist, however. We will have to wait until ongoing cohorts like the Park West and PPMI have accrued a sufficient number of brain samples.

Comment-4

(4) In many cases, It is not clear for different comparisons how the authors adjusted the p-value for the multiple testing

Response-4:

We apologize for not being sufficiently clear on this important point. We have now specified this in the methods and figures.

Comment-5

(5) We can see there are more males than females in nCI than in CI, which can explain the difference in alcohol consumption and cigarettes

Response-5:

We thank the Reviewer for the insightful comment. We agree and have included this in the Discussion (lines 341-343)

Minor comments

(our answers are shown in italics)

(1) What does mean neurologically healthy controls – individuals affected by other neurological conditions

It means individuals without known neurological disease or pathological evidence of neurological disease.

(2) To avoid using the abbreviations in the title e.g CI (abbreviation also used for Confidence Interval

We only use CI in conjunction with “CI-PD” or “nCI-PD”. We do not think this is likely to be confused with confidence interval, but are happy to change this at the discretion of the editor.

(3) The comparison can not be only qualitative. Please mention in the text for each comparison (1) the directionality (2) the significance (3) the method of adjustment of p-value Bonferroni vs FDR

These elements are now mentioned in the text. If there are still instances where the Reviewer feels more clarification is needed, we will be happy to add this.

Reviewer #3 (Remarks to the Author):

This is an interesting manuscript which provides new information around complex I deficiency in iPD. This is an area where research is challenging because of the use of post-mortem tissue and the heterogeneity between patients and even between controls. Use of human tissue is essential since animal models do not help with these age related changes.

I have the following comments:

Comment-1:

1. The authors have used two well characterised cohorts of patients for their studies validating their results which is I think optimal for these sort of studies. The fact that they see clear evidence of complex I deficiency in both cohorts is important

Response-1:

We thank the Reviewer for the positive evaluation of our study.

Comment-2:

2. I am somewhat concerned about their use of only immunohistochemistry for defining complex I deficiency, and indeed evidence of changes in other complexes. I know the authors have used this technique for other studies but there are more quantitative ways of measuring protein abundance in individual cells using fluorescent antibodies and image analysis software. Use of these techniques would allow a more accurate assessment of complex I (or other deficiencies). At least I think they have to better justify their approach or preferably it would be nice to see a limited number of cases assessed using more quantitative techniques

Response-2:

We thank the Reviewer for this suggestion. The reason we resorted to this method, rather than fluorescent staining and signal quantification is that we wanted to rely on a robust, simple, and highly reproducible (even if less sensitive) measure of complex I deficiency which is dichotomous (i.e., positive or negative) and not prone to technical and quantification artifacts/noise.

Quantification of immunofluorescence is certainly possible and there are several published examples of appropriate use of this method on both PD and mitochondrial disease. It does, however, also suffer from certain limitations, including susceptibility to technical variation and artifacts, as well as issues with normalization. For instance, the correct interpretation of complex I immunofluorescence signal requires appropriate normalization for total mitochondrial mass. While this can be achieved using a mitochondrial marker such as VDAC1, it requires that both the CI and VDAC stainings are of perfectly consistent intensity both within each section and across all samples. In our experience this is often not the case, resulting in over- and underestimation of neuronal CI content due to small inconsistencies in the staining intensity. In contrast, a dichotomous evaluation of positive and negative (i.e., similar to the negative control) neurons, while less sensitive, is robust to small interindividual variations in staining intensity and less prone to technical interindividual variation.

Since the clustering of the samples was a critical step for this work, we opted to do this using the proportion of neurons staining negative (or positive) for CI. This approach may be less sensitive, since it misses intermediates and weak cells, but is highly specific and reproducible, as evident by the high interrater reliability (now added in Fig 1) and the replicability across two entirely independent cohorts.

Having said that, we acknowledge that the classification should not rely on a single measure. To increase the robustness of our classification we have resorted to supporting findings from entirely different measures, i.e., not MRC quantifications, showing that the proposed subtypes are characterized by:

- a) Distinct transcriptomic profile in bulk tissue
- b) Distinct cell type-specific gene expression profile, shown by single-nucleus RNA sequencing (entirely new analyses)
- c) Different mtDNA profiles: increased load of neuronal mtDNA deletions in CI-PD
- d) Different clinical features, a predilection for non-tremor dominant motor phenotypes in CI-PD.

Comment-3:

3. Their analysis of single cell mtDNA is interesting especially using the PFC – the difficulty is that they cannot directly look at those neurons which are complex I deficient. They are correct in that we do not know whether the deletion causes the deficiency or the complex I deficiency causes the

deletion. This raises the interesting question, which I do not think is tackled in the manuscript, is what is causing the complex I deficiency. I know not a central point of the manuscript but interesting to see the authors speculations.

Response-3:

We fully agree with the Reviewer that, ideally, we should be able to provide mtDNA data from CI-positive and CI-negative neurons. We have indeed attempted to perform these analyses, but they have proven technically very challenging, as we are unable to achieve good enough multi-fluorescent staining quality in thick frozen sections to be able to confidently discern and micro-dissect CI positive and negative neurons. We are currently improving the methodology, solving technical limitations, and we hope to be able to conduct and publish these analyses in a future paper. We have added a section to the Discussion speculating on the causes of complex I deficiency and the potential role of the mtDNA.

Comment-4:

4. Another point worth the authors discussing is around the mechanism by which the complex I deficiency could be affecting the phenotype of iPD? Is the deficiency seen in different areas of the brain responsible?

Response-4:

This is an intriguing question. In CI-PD, CI deficiency occurred in most of the brain regions we examined, including the prefrontal, cingulate, temporal, and occipital cortex, SNc, CA1 region of the hippocampus, amygdala, and striatum (see Fig 3 in the manuscript). These regions have multiple and diverse roles on motor and non-motor function, and many of them are known to have a role in the phenotypical variance of PD.

The main phenotypical difference between CI-PD and nCI-PD was that the first was strongly correlated with a non-tremor dominant phenotype (i.e., akinetic rigid, or PIGD). This phenotype is generally more severe than the tremor-dominant phenotype, being characterized by faster progression, more debilitating motor dysfunction, and poorer treatment outcomes, as well as more severe and widespread neurodegenerative changes.

It is possible that the observed association between CI-PD and a non-tremor-dominant phenotype is related to a more widespread and severe neuronal dysfunction and loss in CI-PD, induced by the CI deficiency. This is corroborated by the bulk RNAseq data indicating more neuronal loss in CI-PD. We have added this to the Discussion (lines 325-330)

Comment-5:

5. The authors make a very good point about the heterogeneity of complex I deficiency is important if considering clinical trials and their results. The authors suggest that agents might need to be targeted to those where there is the most evidence of complex I deficiency which is true. However it is important that the authors recognise that at present there is no effective treatment for complex I deficiency.

Response-5:

We agree with the reviewer and have updated the text in the discussion to reflect this (lines 334-336)

REVIEWER COMMENTS

Reviewer #1 (Remarks to the Author):

1. The authors have rewritten the introduction, and it is improved. My previous comment regarding the etiological heterogeneity of Parkinson's was intended to support my recommendation for putting the mitochondrial pathway in a broader context, not to imply the authors were generating a novel hypothesis.
2. The data are much improved by the addition of staining of an additional four CxI subunits.
3. I agree that respiratory chain activity declines with age, although this is not selective for CxI. I note the additional information on the brain samples, and the observation that the PMI was not associated with the severity of CI deficiency in either the SN or PFC is reassuring.
4. I accept the authors' explanation. On a related note, it is interesting to speculate that the CxI deficiency is also present in glia in the CxI-PD group.
5. Noted. The authors may wish to revisit their referencing, their new ref 21 is better suited to be with their refs 11-13. There are several papers on platelet mitochondrial function in Parkinson's, including Ann Neurol 1992; 32:782-788 and Brain 1993; 116: 1451-1463. These and others show a significant spread of CxI activity in Parkinson's that would support a proportion being true CxI-related.
6. Noted.
7. Noted. The distinct cell type-specific gene expression profile between the CxI-PD and nCxI-PD groups is interesting.
8. On a separate note, the authors should not refer to CxI activity results (they emphasize that this was not measured) in the manuscript but rather CxI levels, this better reflects their findings on immunohistochemistry.

Reviewer #2 (Remarks to the Author):

I reviewed this study last year, I focused my review on the sections I discussed with the authors previously and the new parts.

- 1) Despite the authors' arguments, the clustering analyses are not robust.
 - 1.1) The authors mention determining clusters visually, but there's no visual plot or statistical test provided to support this cluster analysis. The distribution shown in Fig 1g, particularly the CI, seems like an outlier, suggesting k-means clustering may be unsuitable.
 - 1.2) It is unclear for me about the choice of the k-means method. K-means is a multivariate method, which may not be the best choice for this data.
 - 1.3) What is the rationale behind transforming a continuous variable into a categorical variable? This risks reducing the study's power, especially given its limited size. The authors might consider performing

regression analyses instead of using this subgrouping system.

1.4) I'm uncertain about the purpose of Fig 1C in relation to the clustering.

1.5) The statement "does not involve learning algorithms" seems ambiguous. If that's not the case, then the study's claim of ultimately finding two PD subtypes determined visually becomes harder to understand.

1.6) They state, "We first clustered each of the Norwegian and Spanish cohorts independently."

However, the text contradicts this by saying in the beginning, "Individuals with iPD were classified by k-means clustering of the entire study..."

2. Validation of subtype in others cohorts: Given the authors' own acknowledgment of the weakness in their cluster definition, stating, "Although we defined groups of iPD, the exact threshold between them remains arbitrary," I find it hard to see why there's a robust claim for their existence without further validation.

3. It is not a single assay in the sense that the authors cannot determine whether the neurons with more CI deficiencies and mtDNA deletions are related. As a result, they cannot establish a causal relationship between the two.

4. The authors suggest that incorporating this classification into clinical practice may be a crucial step in precision medicine. However, at this stage, I don't know how: the clinical features, TD, and alcohol demonstrate only a weak correlation with this division. It's unclear how much they contribute to building a model that predicts nCI versus CI at the individual level and thereby identifies both subtypes. This is where my primary concern lies: a deeply longitudinal phenotyped study (and genetically characterized) would be useful in extracting the main features that predict both subtypes.

5. NA

6. Why not adjust for gender and see if it affects the differences between groups regarding alcohol consumption?

The majority of the signals from their bulk analyses arise from cell heterogeneity. They observed a decrease in GABA intergenic neurons in CI-PD. The authors are conducting an additional single-cell study in a new manuscript. However, I have two concerns:

This study doesn't address the issue that I raised previously.

While the idea of gaining new insights is commendable, there is no in-depth analysis of this SC dataset. The authors provide the number of DE genes by cell type, but which genes are upregulated and which are downregulated? What pathways are involved? Are any related to mitochondrial dysfunction? Did they find differences in terms of population in GABA intergenic neurons between both subtypes?

I'm convinced that this dataset could offer valuable insights when analysed in relation to other single-cell data and PD genetics but this dataset should be the central point of a study

Reviewer #3 (Remarks to the Author):

The authors have responded to my comments:

1. they give a reasonable explanation as to why they used immunohistochemistry rather than immunofluorescence - they have added inter-rater variability which I think strengthens their approach
2. I appreciate the methodological challenges in obtaining good immunofluorescent staining on thick enough section to extract DNA from single cells. I do not think it is necessary to have this data for this paper
3. the authors have responded appropriately to my comment on on the effect of complex I deficiency on phenotype

Reviewer #1 (Remarks to the Author):

Comments 1:

The authors have rewritten the introduction, and it is improved. My previous comment regarding the etiological heterogeneity of Parkinson's was intended to support my recommendation for putting the mitochondrial pathway in a broader context, not to imply the authors were generating a novel hypothesis.

Response-1:

We thank the reviewer for clarifying this and for the positive comment regarding our introduction.

Comment-2:

The data are much improved by the addition of staining of an additional four CxI subunits.

Response-2-2:

We thank the reviewer for this evaluation.

Comment-3-2:

I agree that respiratory chain activity declines with age, although this is not selective for CxI. I note the additional information on the brain samples, and the observation that the PMI was not associated with the severity of CI deficiency in either the SN or PFC is reassuring.

Response-3-2:

We thank the reviewer for their comment.

Comment-4-2:

I accept the authors' explanation. On a related note, it is interesting to speculate that the CxI deficiency is also present in glia in the CxI-PD group.

Response-4-2:

We fully agree that the glial involvement is a relevant question, and we are currently conducting a separate study to explore this question.

Comment-5:

Noted. The authors may wish to revisit their referencing, their new ref 21 is better suited to be with their refs 11-13. There are several papers on platelet mitochondrial function in Parkinson's, including Ann Neurol 1992; 32:782-788 and Brain 1993; 116: 1451-1463. These and others show a

significant spread of CxI activity in Parkinson's that would support a proportion being true CxI-related.

Response-5:

We agree with the reviewer and have followed their suggestions: we have included the original Schapira et al. 1989 reference together with refs 11-13. Furthermore, we have included a section on the platelet findings and appropriate references (e.g., Benecke et al Brain 1993; 116: 1451-1463, Yoshino et al; Neural Transm Park Dis Dement Sect . 1992;4(1):27-34). Please see page 5, lines 99-101.

Comment-6:

Noted.

Response-6:

We thank the reviewer for acknowledging our response.

Comment-7:

Noted. The distinct cell type-specific gene expression profile between the CxI-PD and nCxI-PD groups is interesting.

Response-7-2:

We thank the reviewer for their comment. We have further expanded on the differences in cell type-specific gene expression between the CI-PD and nCI-PD proposed subtypes and are currently pursuing further research on this matter.

Comment-8-1:

On a separate note, the authors should not refer to CxI activity results (they emphasize that this was not measured) in the manuscript but rather CxI levels, this better reflects their findings on immunohistochemistry.

Response 8-1:

We thank the reviewer for pointing this out and have changed it accordingly in the text.

Reviewer #2 (Remarks to the Author):

Comment

1) Despite the authors' arguments, the clustering analyses are not robust.

1.1) The authors mention determining clusters visually, but there's no visual plot or statistical test provided to support this cluster analysis. The distribution shown in Fig 1g, particularly the CI, seems like an outlier, suggesting k-means clustering may be unsuitable.

1.2) It is unclear for me about the choice of the k-means method. K-means is a multivariate method, which may not be the best choice for this data.

Response

To clarify, we did not determine the clusters visually. Clustering of the samples was done by k-means and supported by multiple other methodologies (see our detailed response below). What we determined visually was the number of clusters (i.e., before clustering the samples). This is a reasonable approach when founded in the biological question at hand. It is clear from mere observation that the PD group has a much higher variance/spreading than the controls, with some patients overlapping with the controls and some being away from the controls. Whether the PD group should be divided into 2 or 3 clusters is arbitrary and depends on the question asked. We believe that differentiating the CI-PD into two parts (i.e., severe and mild) may hold additional information. However, the chosen number of clusters is also supported by ANOVA analyses of the clustered data, with 3 clusters having the highest F score.

Number of clusters	F-statistics (ANOVA) of clustered data
2	339
3	450
4	317

The choice of k-means was driven by the fact that it is the most widely-used unsupervised clustering algorithm and does not require parametrization. The reviewer is correct in that the k-means algorithm is a multivariate method, and in fact it can be seen as a generalization of the univariate Jenks natural breaks optimization (with a difference in how the group distances are minimized). Using the Jenks natural breaks optimization instead, results in nearly identical results, with only a single sample being assigned to a different cluster (Fig. 1).

Fig. 1 Clusters based on Jenks natural breaks. Clustering of all subjects based on the proportion of CI positive neurons in the prefrontal cortex (y-axis) according to Jenks natural breaks results in three clusters (red: cluster_1, green: cluster_2, blue: cluster_3). The x-axis shows the clusters according to our original K-means clustering. Compared with clustering according to k-means, only a single subject changes group from the nCI-PD group to mild CI-PD (blue dot in the green cluster).

Other clustering alternatives are also yielding similar results. Non-parametric density estimation¹ results in 3 clusters, with very similar assignments (only 4 samples are assigned to a different cluster, Fig. 2).

Fig. 2. Clusters according to non-parametric density estimation. Clustering of all subjects based on the proportion of CI positive neurons in the prefrontal cortex (y-axis) according to non-parametric density estimation results in three clusters (colors). The x-axis correspond to our original K-means clustering. Compared with clustering according to k-means, four subjects change group from the nCI-PD (cluster_3) group to mild CI-PD (cluster_2).

Since values are fractional, we also clustered the samples based on a mixture of beta distributions. BIC and AIC values suggested 2 clusters in this case (Fig. 3).

Figure 3. BIC and AIC levels based on a mixture of beta distributions. The AIC and BIC values (y-axis) of models based on a mixture of beta distributions suggest two clusters (x-axis).

Samples from the two k-means clusters corresponding to the lowest CI-positivity constitute a single cluster, but otherwise the 2-group classification is very similar (Fig. 4)

Fig. 4. Clusters according to beta distributions Clustering of all subjects based on the proportion of CI positive neurons in the prefrontal cortex (y-axis) according to beta distributions results in two clusters (colors). Cluster 1 contains both the mild and severe CI-PD groups (cluster_1 and cluster_2) and cluster away from the controls. Cluster 2 contain all the controls and correspond to the nCI-PD group. Compared with clustering according to k-means (x-axis), two subjects changes group from the nCI-PD (cluster_3) group to mild CI-PD (cluster_2).

As shown above, four different clustering approaches yield highly similar results. Seeing a few samples switch group with different approaches is expected. The main point of our findings is that neuronal CI deficiency is not a pervasive feature of iPD, but one that occurs in a subgroup of individuals, while the rest are in the range of healthy controls. We do not claim to determine an exact numerical threshold for each group. As with many classification approaches in medicine and biology, there will always be borderline cases which change group according to which stratification method is used.

Comment

1.3) What is the rationale behind transforming a continuous variable into a categorical variable?

This risks reducing the study's power, especially given its limited size. The authors might consider performing regression analyses instead of using this subgrouping system.

Response

The rationale for using a categorical classification is a biological one. While it is true that % CI-positive neurons is a continuous variable, it is clear from our data that most of the iPD population overlaps with healthy controls. Thus, from the medical, disease-perspective it is interesting to identify the features associated specifically with the iPD subgroup that deviates from the controls (i.e., that has higher levels of neuronal CI deficiency). In fact, despite the potential reduction of power of discovery from using a categorical variable, we have substantial signal, supporting our hypothesis of two distinct iPD groups based on the presence of CI deficiency.

Nevertheless, considering the Reviewer's comment, we have performed a linear regression to evaluate the relationship between % CI positive neurons and the clinical variables in our dataset.

First, we explored the data in univariate models using % CI-positivity as the response variable:

Predictor variable	Adjusted R ²	Beta	P-value
Age of first reported symptom	0.002	0.06	0.286
Age of death	0.015	0.11	0.127
Disease duration	-0.013	-0.01	0.951
Total UPDRS final examination	-0.004	-0.06	0.364
Ethanol units	0.096	0.36	0.031
Daily cigarettes	0.001	0.13	0.316
Sex	0.025	-2.01	0.075
Dementia	-0.008	-0.68	0.554
Phenotype AR/TD	0.050	-2.85	0.051
Phenotype PIGD/TD	0.245	-6.20	0.002

The following predictor variables showed significant level of correlation:

- Age of death and age of first reported symptom ($r(82) = 0.692$, $P = 2.23 \times 10^{-13}$)
- Age of first reported symptom and disease duration ($r(83) = 0.458$, $P = 2.39 \times 10^{-14}$)
- Phenotype AR/TD and phenotype PIGD/TD (Fisher's exact test, $P = 0.002$)
- Total UPDRS final examination and dementia (Point-Biserial correlation: $r(38) = 0.363$, $P = 0.023$)
- Total UPDRS final examination and phenotype PIGD/TD (Point-Biserial correlation: $r(32) = 0.492$, $P = 0.004$)
- Dementia and ethanol units (Point-Biserial correlation: $r(39) = -0.344$, $P = 0.030$)

The majority of the above correlations are expected.

Notably, there was no correlation between sex and daily cigarettes (Point-Biserial correlation: $r(39) = -0.275$, $P = 0.086$) or sex and ethanol units (Point-Biserial correlation: $r(39) = -0.067$, $P = 0.681$).

To specifically test whether sex and ethanol units were confounding variables, these were run in a separate model, finding that sex does not predict the weekly number of ethanol units (Beta = -0.51, Adjusted R² = -0.02, $P = 0.681$) in our cohort.

Considering the multi-collinearity of the predictor variables, the significance levels from the univariate analysis and the low number of subjects in the phenotype PIGD/TD and ethanol units

variables (n = 40), we further chose the following predictor variables for the multivariate linear regression: sex, phenotype PIGD/TD and ethanol units. Cases were excluded listwise.

The overall regression was statistically significant (n = 33, adjusted $R^2 = 0.380$, $P = 7.1 \times 10^{-4}$). Motor phenotype PIGD/TD (Beta = -6.29, $P = 8.2 \times 10^{-4}$) and sex (Beta = -3.76, $P = 0.036$) significantly predicted the % CI positivity. Ethanol units (Beta = 0.41, $P = 0.072$) did not significantly predict the % CI positivity in this model.

In conclusion, while we agree with the reviewer that we risk losing power in categorizing our data, repeating the analyses using a linear model produces highly similar results while still facing the potential loss of power and risk of overfitting our model due to a limited sample size.

Comment

1.4) I'm uncertain about the purpose of Fig 1C in relation to the clustering.

Response

1.4) We thank the reviewer for this comment and see that our previous response may not have been sufficiently clear on the purpose of Fig 1C. Since the % CI positive neurons was assessed by two independent raters, we assess inter-rater reliability. The figure shows the Bland-Altman plot with the difference between the measurements of the two observers (O1 and O2). Like the reviewer correctly points out, the fact that there was excellent inter-rater agreement did not impact our choice of clustering method. The point of mentioning the inter-rater agreement in our previous response was to emphasize that the robustness of the data has improved since first submission.

Comment

1.5) The statement "does not involve learning algorithms" seems ambiguous. If that's not the case, then the study's claim of ultimately finding two PD subtypes determined visually becomes harder to understand.

Response

We do not fully grasp the Reviewer's point. As indicated above, the existence of the subtypes is supported by multiple lines of evidence and replicated independently in two different populations. What we meant by "*does not involve learning algorithms*" in our previous response was merely the fact that we did not employ machine learning to classify our samples. See also our reply to Comment-2.

Comment

1.6) They state, "We first clustered each of the Norwegian and Spanish cohorts independently." However, the text contradicts this by saying in the beginning, "Individuals with iPD were classified by k-means clustering of the entire study..."

Response

We thank the Reviewer for spotting this error in our text. The text and Fig 1g were not correct and have been updated. Each of the cohorts was indeed clustered independently before the combined dataset was clustered. The clustering in the combined cohort was consistent with the clustering based on the separate cohorts, with only one individual from the Spanish cohort (ESP) changing groups from nCI-PD to mild CI-PD after clustering of the combined cohorts.

Comment

2) Validation of subtype in others cohorts: Given the authors' own acknowledgment of the weakness in their cluster definition, stating, "Although we defined groups of iPD, the exact threshold between them remains arbitrary," I find it hard to see why there's a robust claim for their existence without further validation.

Response

As stated above (see our responses to Comments1, 1.1 and 1.2), the existence of the subtypes is supported by multiple lines of evidence and replicated independently in two different populations. While we are presenting robust evidence supporting the existence of the CI-PD and nCI-PD subtypes, this does not mean that we should be able to determine an exact and clearcut threshold between the groups in the form of a specific number. As with many classification approaches in medicine and biology, there will always be borderline cases which may change group according to which stratification method is used.

The main point of our findings is that neuronal CI deficiency is not a pervasive feature of iPD, but one that occurs in a subgroup of ~25% of the iPD population, while the rest are in the range of healthy controls. We believe that this is robustly shown in our data. We do not claim to determine an exact numerical threshold separating the groups, nor was this the purpose of our study (see also our response to comments 1.1-1.2).

Furthermore, looking at earlier literature, as the Reviewer asked to do in a previous comment, strongly supports the existence of these subtypes (see our response to Reviewer-1, Comment 5). The individual data from these studies are not available to us to be able to run similar clustering approaches, but a simple inspection of the data, clearly shows that they are in line with our findings. We have replicated our findings in two independent cohorts. Further replication and validation in other cohorts and by other groups will of course be needed, as with any discovery. However, we respectfully believe that our findings are of sufficient robustness and quality to warrant publication at this stage, so that others may have the opportunity to perform further validation.

We would also like to stress that there are currently extremely few cohorts in the world with both postmortem brain tissue and systematic, prospectively clinical data, which can be used to validate

our findings. It is therefore important that our results now are shared with the community so that any further validation in the few other available samples will be possible.

Comment

3) It is not a single assay in the sense that the authors cannot determine whether the neurons with more CI deficiencies and mtDNA deletions are related. As a result, they cannot establish a causal relationship between the two.

Response

We fully agree with the Reviewer that, ideally, we should be able to provide mtDNA data from CI-positive and CI-negative neurons. We have indeed attempted to perform these analyses, but they have proven technically very challenging, as we are unable to achieve good enough multi-fluorescent staining quality in thick frozen sections to be able to confidently discern and micro-dissect CI positive and negative neurons. We are currently improving the methodology, solving technical limitations, and we hope to be able to conduct and publish these analyses in a future paper. We have added a section to the Discussion speculating on the causes of complex I deficiency and the potential role of the mtDNA.

Comment-3

(3) Both cohorts are not clinically characterised enough (unlike PPMI) to decide which clinical feature is the most relevant for each subtype.

Response-3:

The Norwegian cohort has excellent clinical data, as these samples originate from the ParkWest cohort, a prospective, longitudinal and systematically characterized cohort, which has been described in detail². The Spanish cohort is from the Barcelona Brain Bank and, like most brain banks, lacks prospectively collected systematic clinical information.

While we agree with the Reviewer that, in an ideal world, we would have prospective and systematic clinical information on both cohorts, it is important to consider what is currently realistic. There are presently very few cohorts with both prospective systematic clinical data and brain collection in adequate numbers. Park West is one of those rare and extremely valuable cohorts, which has accumulated a relatively large number of brains, because it has been going for almost 20 years (since 2004). There is no doubt that PPMI has excellent clinical information, but we believe that the comparison made by the Reviewer is somewhat unfair, since the PPMI has not yet accumulated a sufficiently large brain collection and is would, therefore, not have been useful in the present study.

In our data, we see a strong association between CI-PD and akinetic rigid and/or PIGD phenotype and this is replicated in both cohorts. We believe that this is already important novel insight into the

clinical correlates of the CI-PD subtype. That said, observations in much larger cohorts (several hundreds) will be necessary to identify the full clinical phenotype associated with CI-PD and nCI-PD. Such cohorts, combining prospective, systematic clinical data and brain collection from hundreds of individuals with PD and controls do not currently exist, however. We will have to wait until ongoing cohorts like the Park West and PPMI have accrued a sufficient number of brain samples.

Comment-3-2:

The authors suggest that incorporating this classification into clinical practice may be a crucial step in precision medicine. However, at this stage, I don't know how: the clinical features, TD, and alcohol demonstrate only a weak correlation with this division. It's unclear how much they contribute to building a model that predicts nCI versus CI at the individual level and thereby identifies both subtypes. This is where my primary concern lies: a deeply longitudinal phenotyped study (and genetically characterized) would be useful in extracting the main features that predict both subtypes.

Response-3-2:

We agree with the reviewer that implementation of these findings in clinical practice will not be trivial. As mentioned in our discussion (page 20, lines 491-499), accurate and impactful implementation of the CI-PD and nCI-PD classification in the clinic will require the emergence of peripherally applicable biomarkers that can identify the subtypes in vivo. We are, in fact, currently working on this, looking at multiple tissues from our prospective cohort, including muscle, platelets, blood, and CSF. As previous findings have indicated (see our response to Reviewer-1, comment 5) and our current ongoing work corroborates, several of these tissues do, indeed, generate evidence supporting the presence of a CI-PD subtype which accounts for ~25% of the cases. However, proving that the peripheral findings reflect the CI-PD subtype in the patient brain will require examining brain tissue from the same individuals from whom muscle and platelets have been examined. This is not yet available to us or, to the best of our knowledge, in any other cohort. In time, when these brains become available, the specificity and sensitivity of the peripheral, clinically applicable biomarkers can be assessed.

We fully agree with the Reviewer that a large, deeply longitudinally phenotyped study that is genetically characterized would be very useful in extracting the main features that predict both subtypes. However, such cohorts are exceedingly rare. Apart from our Park West cohort, which fulfills all of these conditions, very few other samples with both brain tissue and systematic, prospectively collected clinical data exist,

We hope that large ongoing cohorts, such as the PPMI and our own cohorts, will one day address this need. However, these cohorts have not yet collected postmortem brain tissue (at least not in numbers that would be of use here).

Comment-4

(4) In many cases, It is not clear for different comparisons how the authors adjusted the p-value for the multiple testing

Response-4:

We apologize for not being sufficiently clear on this important point. We have now specified this in the methods and figures.

Comment-4-2:

NA

Comment-5

(5) We can see there are more males than females in nCI than in CI, which can explain the difference in alcohol consumption and cigarettes

Response-5:

We thank the Reviewer for the insightful comment. We agree and have included this in the Discussion (lines 341-343)

Comment-5-2:

Why not adjust for gender and see if it affects the differences between groups regarding alcohol consumption?

Response-5-2:

We agree with the reviewer that this could be of value and have performed regression analyses on our data. Please see our answer 1.3) to comment 1.3. Sex is a predictor of % CI positivity in our multivariate model but does not seem to confound the total number of weekly ethanol units alone. When phenotype PIGD/TD and ethanol consumption is fitted in the model, the effect of ethanol consumption alone is lost. Thus, phenotype PIGD/TD may confound the total number of weekly ethanol units.

Comment-6:

The majority of the signals from their bulk analyses arise from cell heterogeneity. They observed a decrease in GABA intergenic neurons in CI-PD. The authors are conducting an additional single-cell study in a new manuscript. However, I have two concerns:

This study doesn't address the issue that I raised previously.

While the idea of gaining new insights is commendable, there is no in-depth analysis of this SC dataset. The authors provide the number of DE genes by cell type, but which genes are upregulated and which are downregulated?

What pathways are involved? Are any related to mitochondrial dysfunction?

Did they find differences in terms of population in GABA intergenic neurons between both subtypes?

I'm convinced that this dataset could offer valuable insights when analysed in relation to other single-cell data and PD genetics but this dataset should be the central point of a study

Response-6:

We thank the reviewer for this comment and have performed more in-depth analyses of the snRNAseq dataset to provide additional insight. Sections describing and interpreting the new analyses and data have been added to the Results (page 14, lines 322-353) and Discussion (page 19, lines 457-469), as well as in the Methods (page 33, lines 804-808), Supplementary Table 11 and Supplementary Figures S10-S13.

All differentially expressed genes for all cell types are provided in Supplementary Table 09 (CI-PD vs controls), Supplementary Table 10 (nCI-PD vs. controls), and Supplementary Table 11 (CI-PD vs. nCI-PD). In each table, the coef (column B) shows the log₂ fold change of each gene. A negative coef indicates that the gene is downregulated, while a positive coef indicates that the gene is upregulated. Due to the large number of differentially expressed genes, we chose to refrain from commenting on individual genes with few notable exceptions, including genes with a known link to PD and/or mitochondrial function, or genes found to be differentially expressed across multiple cell types. For instance, we report that nuclear genes encoding MRC proteins are significantly downregulated in CI-PD compared to nCI-PD.

Pathway-enrichment analyses have limited value in single-nucleus datasets due to the relatively low number and highly cell-specific nature of the genes captured. However, with this limitation in mind, we have performed functional enrichment analyses in the most information-rich contrast (i.e., CI-PD vs nCI-PD). These analyses show that the differentially expressed genes between CI-PD and nCI-PD are significantly enriched for pathways relating to the mitochondrial respiratory chain and OXPHOS.

In terms of cell counts, the snRNAseq dataset indicated no significant differences in the proportions of cell types between CI-PD vs nCI-PD. This is, however, not surprising. Nuclei counts in snRNAseq datasets cannot be considered representative of cell proportions in the tissue, due to potential selective losses or gains along the process of nuclei isolation and processing. In that regard, the bulk-tissue based estimates are more reliable, albeit indirect estimates

As the reviewer points out, this is a complex and multilayered dataset, which should be the central point of a study. We are indeed conducting more in-depth computational experiments with this dataset which, we hope, will form the basis for more findings in the near future. However, in the context of the current paper which focuses on demonstrating the existence of the CI-PD and nCI-PD subtypes, we respectfully believe that these expanded analyses are sufficient to robustly support our conclusions.

Reviewer #3 (Remarks to the Author):

The authors have responded to my comments:

1. they give a reasonable explanation as to why they used immunohistochemistry rather than immunofluorescence - they have added inter-rater variability which I think strengthens their approach
2. I appreciate the methodological challenges in obtaining good immunofluorescent staining on thick enough section to extract DNA from single cells. I do not think it is necessary to have this data for this paper
3. the authors have responded appropriately to my comment on on the effect of complex I deficiency on phenotype

Response-6:

We thank the reviewer for their comments and for the positive evaluation of our manuscript.

REVIEWERS' COMMENTS

Reviewer #2 (Remarks to the Author):

Although I still have concerns regarding the direct actionable mechanistic understanding and treatment for Parkinson's disease, particularly considering the use of postmortem brain tissue, I acknowledge that the authors have carefully addressed all of my comments.

Major Point: Could the authors include references to public repositories in the manuscript for their transcriptomics datasets, encompassing both bulk and RNA data? With the advancement of single-cell datasets derived from postmortem brain tissue, these could serve as valuable resources for replicating their findings.

Minor point:

The F-statistic from ANOVA is not the conventional method for determining the optimal number of clusters in K-means clustering analysis. Instead, methods such as the Elbow Method are more typically employed. There are other, more suitable approaches for performing clustering with a single variable

Reviewer #2 (Remarks to the Author):

Comment 1:

Although I still have concerns regarding the direct actionable mechanistic understanding and treatment for Parkinson's disease, particularly considering the use of postmortem brain tissue, I acknowledge that the authors have carefully addressed all of my comments.

Response 1:

We thank the reviewer for acknowledging our response to their comments.

Comment 2:

Major Point: Could the authors include references to public repositories in the manuscript for their transcriptomics datasets, encompassing both bulk and RNA data? With the advancement of single-cell datasets derived from postmortem brain tissue, these could serve as valuable resources for replicating their findings.

Response 2:

The reference to the public repository for the single nucleus RNA dataset has been added to the manuscript (page 35, lines 840-842 and 845-846). The reference to the public repository for the bulk transcriptomic data is given in the manuscript (page 35, lines 840-842 and 845-846).

Comment 3:

Minor point:

The F-statistic from ANOVA is not the conventional method for determining the optimal number of clusters in K-means clustering analysis. Instead, methods such as the Elbow Method are more typically employed. There are other, more suitable approaches for performing clustering with a single variable

Response 3:

We appreciate the feedback from the reviewer. We concur that the F-statistic from ANOVA is not the most conventional method for determining the optimal number of clusters in K-means clustering analysis. As we have noted in our previous response, we employed multiple methods to ascertain the optimal number of clusters, with the results indicating a range of 2-4 clusters depending on the method utilized. Ultimately, after visual inspection of the data and considering the support from various other clustering methods, as well as the biological context of this classification, we decided on three clusters as the most appropriate for our analysis.